# Cytokines and chemokines profile in encephalitis patients: A meta-analysis

**Alireza Soltani Khaboushan**[1,2,3], **Mohammad-Taha Pahlevan-Fallahy**[1,2,3], **Parnian Shobeiri**[1,3,4], **Antônio L. Teixeira**[5], **Nima Rezaei**[3,6]\*

**1** School of Medicine, Tehran University of Medical Sciences, Tehran, Iran, **2** Students' Scientific Research Center, Tehran University of Medical Sciences, Tehran, Iran, **3** Systematic Review and Meta-Analysis Expert Group (SRMEG), Universal Scientific Education and Research Network (USERN), Tehran, Iran, **4** Non–Communicable Diseases Research Center, Endocrinology and Metabolism Population Sciences Institute, Tehran University of Medical Sciences, Tehran, Iran, **5** Neuropsychiatry Program, Department of Psychiatry and Behavioral Sciences, McGovern Medical School, The University of Texas Health Science Center at Houston, Houston, TX, United States of America, **6** Department of Immunology, School of Medicine, Tehran University of Medical Sciences, Tehran, Iran

\* rezaei_nima@yahoo.com, Rezaei_nima@tums.ac.ir

**Data Availability Statement:** All relevant data are within the manuscript and its Supporting Information files.

## Abstract

### Background

Encephalitis is caused by autoimmune or infectious agents marked by brain inflammation. Investigations have reported altered concentrations of the cytokines in encephalitis. This study was conducted to determine the relationship between encephalitis and alterations of cytokine levels in cerebrospinal fluid (CSF) and serum.

### Methods

We found possibly suitable studies by searching PubMed, Embase, Scopus, and Web of Science, systematically from inception to August 2021. 23 articles were included in the meta-analysis. To investigate sources of heterogeneity, subgroup analysis and sensitivity analysis were conducted. The protocol of the study has been registered in PROSPERO with a registration ID of CRD42021289298.

### Results

A total of 23 met our eligibility criteria to be included in the meta-analysis. A total of 12 cytokines were included in the meta-analysis of CSF concentration. Moreover, 5 cytokines were also included in the serum/plasma concentration meta-analysis. According to the analyses, patients with encephalitis had higher CSF amounts of IL-6, IL-8, IL-10, CXCL10, and TNF-α than healthy controls. The alteration in the concentration of IL-2, IL-4, IL-17, CCL2, CXCL9, CXCL13, and IFN-γ was not significant. In addition, the serum/plasma levels of the TNF-α were increased in encephalitis patients, but serum/plasma concentration of the IL-6, IL-10, CXCL10, and CXCL13 remained unchanged.

**Funding:** The authors received no specific funding for this work.

**Competing interests:** The authors have declared that no competing interests exist.

**Abbreviations:** NMDAR, N-Methyl-D-Aspartate Receptor; MRI, magnetic resonance imaging; CSF, cerebrospinal fluid; EEG, electroencephalograms; IL-1, Interleukin-1; EEG, electroencephalogram; IL-1, interleukin-1; IFN-γ, interferon-γ; TNF-α, tumor necrosis factor-α; SD, standard deviation; IQR, interquartile range; JBI, Joanna Briggs Institute's; ES, effect size; CI, confidence interval; ELISA, enzyme-linked immunosorbent assay; HSV-1, Herpes simplex virus; HSE, herpes simplex encephalitis; CNS, central nervous system; PRR, pattern recognition receptor; TLR, toll-like receptors; IRF3, interferon regulatory factors; pDC, plasmacytoid dendritic cells; NK, Natural Killer; PAMP, pathogen-associated molecular patterns; DEP, differentially expressed proteins; MIP-1, Macrophage inflammatory protein; sTNF-R, soluble tumor necrosis factor receptor-; MDDC, monocyte-derived dendritic cell; JE, Japanese encephalitis; BAFF, B Cell Activating Factor; APRIL, A proliferation-inducing ligand.

## Conclusions

This meta-analysis provides evidence for higher CSF concentrations of IL-6, IL-8, IL-10, CXCL10, and TNF-α in encephalitis patients compared to controls. The diagnostic and prognostic value of these cytokines and chemokines should be investigated in future studies.

## 1. Introduction

Encephalitis is defined as inflammation of active brain tissues that can be caused by autoimmune reactions or infections. Viral pathogens are the main culprit behind infectious encephalitis, including tick-borne encephalitis, Japanese encephalitis, and herpes simplex virus (HSV)-caused encephalitis, among others. More than 100 different pathogens and toxins have been identified as agents contributing to encephalitis pathogenesis [1]. More recently, a growing number of studies have investigated biomarkers for diagnosis and therapeutic targeting of autoimmune encephalitides, such as Anti-N-Methyl-D-Aspartate Receptor (NMDAR) [2–5].

Although the known number of etiologies for encephalitis is on the rise, it is still hard to establish the cause of encephalitis cases in clinical practice, and the underlying mechanisms are complex, not fully understood, and vary according to the cause [6–8]. According to Khetsuriani et al., From 1988 to 1997, about 18,680 cases of encephalitis were hospitalized in the United States each year; with the underlying cause for most (59.5%) not being identified [1]. Patients with encephalitis present with diverse neurological symptoms, including localizing and non-localizing symptoms such as altered mental status and seizures, alongside fever, headache, and meningeal signs [9].

Diagnosing encephalitis and its causative agents is a crucial factor in guiding treatment. A late diagnosis or an incorrect one can lead to negative clinical outcomes. The time of diagnosis might affect the prognosis of the disease vastly. Patients who are diagnosed sooner have a much better prognosis than others [10]. It is of utmost importance to have fast, reliable methods of diagnosing encephalitis in patients in order to improve the clinical outcome and prevent the patients' conditions from getting worse. Currently, the diagnosis of encephalitis is confirmed via neuroimaging methods, mainly brain magnetic resonance imaging (MRI), and measuring cerebrospinal fluid (CSF) biomarkers of central nervous system infection and autoantibodies and interpreting electroencephalograms (EEG) in order to detect specific patterns in brain waves [11, 12]. Some studies have described increased levels of inflammatory biomarkers, such as cytokines and chemokines, in patients with encephalitis [2, 4, 13]. Cytokines and chemokines are essential agents contributing to the regulation of the innate and adaptive immune system and inflammatory reactions. Cytokines are small, low-weight polypeptides secreted by immune cells that play an important role in intercellular interactions. Different cells can secret the same cytokine, and a single cytokine can act on various cell types. They are produced and released in a cascade, leading to increased or decreased inflammatory response based on the cytokine secreted, the target tissue, and the interaction between various biochemical agents [14–18]. Different pro-inflammatory cytokines, such as interleukin (IL)-1, IL-6, IL-12, IL-18, interferon (IFN)-γ, and tumor necrosis factor (TNF)-α and anti-inflammatory ones, such as IL-4, IL-10, IL-13, and IL-19 which are secreted from immune cells interact with body cells to mediate the immune responses in the body and thus have the most optimum inflammatory response [19]. Chemokines are a group of cytokines that are released mainly by leukocytes to induce chemotaxis in damaged tissues and attract white blood cells. In addition,

chemokines could be secreted by tissue-resident cells, such as neurons and glial cells in the brain [20–22]. The term chemotaxis is used to refer to a situation in which cells adjust their movement according to the presence of specific agents in the environment. The reason might be a foreign agent such as a bacterium, fungus, virus, or simply a foreign body. The alteration in the concentration of the different chemokines, such as CXCL10, CXCL-13, CCL-4, CCL-17, CCL-20, and cytokines, including IL-2, IL-4, IL-6, IL-9, IL-10, and IFN-γ have been observed in encephalitis patients [23–30].

This study aims to systematically review studies measuring the levels of interleukins and chemokines in encephalitis, which may help identify and/or develop diagnostic signatures based on these biomarkers. Moreover, these signatures might contribute to a better understanding of the underlying pathophysiology.

## 2. Materials and methods

### 2.1. Search strategy and databases

Potentially eligible studies were found by conducting a systematic search in PubMed, Embase, Scopus, and Web of Science with no date and type of study limits. The comprehensive search string is available as S1 Table. The search has been updated until August 17[th], 2021. The reference list of the retrieved studies has been screened to find further related studies. The Preferred Reporting Items for Systematic reviews and Meta-Analyses (PRISMA) has been used to report the results. The completed PRISMA checklist is available as S1 Checklist. The protocol of the study has been registered in the PROSPERO (https://www.crd.york.ac.uk/prospero/display_record.php?ID=CRD42021289298) with the registration ID of CRD42021289298, and the file is available as S1 Protocol.

### 2.2. Selection criteria

The studies providing the concentration of the cytokines and chemokines in the plasma/serum or CSF have been considered to be eligible for the review. Studies were included if they met the following inclusion criteria: 1) original studies on human subjects, 2) diagnosis of any type of encephalitis, meningoencephalitis, and encephalomyelitis, 3) measurement of the cytokine concentration in the serum, plasma, or CSF of patients and controls (healthy controls or patients without encephalitis and other infectious diseases), and 4) adequate data for calculation of standardized mean difference (SMD). Meanwhile, the studies were excluded if they had any of the following criteria: 1) reviews, book chapters, case reports, meeting abstracts, 2) studies containing animal subjects, 3) in vivo and in vitro studies, 4) studies on gene expression of the cytokines but not their levels, 5) studies that did not have control groups 6) All cancer patients and patients with paraneoplastic encephalitis, because of the existence of cancerous tissue or remote neoplasia that might dysregulate cytokines profile, 7) concurrent complications such as pulmonary edema which could affect the concentration of cytokines, and 8) no measurement of the cytokines in the plasma, serum, or CSF. Screening and eligibility assessment were performed independently by two authors (ASK and MTPF), and discrepancies and conflicts were resolved by discussion. The third author (PS) was consulted for conflict resolution in case of disagreement.

### 2.3. Data extraction

Two authors have extracted the data independently (ASK and MTPF). The following data were extracted from the studies: the first author's name, publication year, location, diagnostic criteria for encephalitis, inclusion and exclusion criteria of patients and controls, sample size,

demographic information (e.g., age and sex), duration of the hospitalization, assay type, sampling sources, type of the cytokine, cytokine concentration, mean and standard deviation (SD) in patients and controls, measurement unit, and WBC levels in blood and CSF, where available. Discrepancies were resolved by discussion and agreement. When enough data was not available in the paper, we contacted the corresponding author to request further data. In case the reported concentration was median and interquartile range (IQR) or range, we used the transformation reported by Wan et al. [31], Luo et al. [32], and Shi et al. [33] to calculate mean and SD.

## 2.4. Quality assessment

The overall quality of the included studies was critically appraised independently by two raters (ASK and MTPF) using the Joanna Briggs Institute's (JBI) checklist for the analytical cross-sectional studies. It has eight questions for quality assessment of the studies [34, 35]. The JBI tool assesses studies based on their inclusion criteria, study subjects and setting, measurement, confounding factors and dealing with them, outcomes, and statistical methods. The list of questions and their detailed definition is available in Table 2. The incongruences in the quality assessment were settled by consulting with the third author (PS).

## 2.5. Data synthesis and meta-analysis

The mean and SD of cytokine levels have been gathered as continuous data in encephalitis patients and the control group. When studies have reported concentrations of cytokines in more than one encephalitis group, if the types of encephalitis were similar (all groups had infectious encephalitis or all groups had autoimmune encephalitis), we pooled the mean and SD of the concentrations in those groups. In case median and IQR or range had been reported instead of mean and SD, they were converted to mean and SD. The between-group Hedges' g standardized mean difference (SMD) was calculated based on sample size, mean, and SD to measure the effect size (ES) of the studies. Hedges' g is similar to Cohen's d with an adjustment for small samples. SMD and its 95% confidence interval (CI) were used to represent the difference in cytokines between the encephalitis group and controls. The ES of 0.2, 0.5, and 0.8 demonstrate a small, moderate, and large effect, respectively [36]. Cochran's Q test was used to assess heterogeneity, and a $P$-value of 0.10 was considered as the existence of heterogeneity. Moreover, $I^2$ was used for a more precise estimation of the heterogeneity, and $I^2 < 25\%$, 25–75%, and >75% are deemed as low, moderate, and high heterogeneity, respectively. Random effect model analysis using the DerSimonian and Laird method was used for meta-analysis, and the $P$-value equal to or less than 0.05 was considered significant [37]. Subgroup analysis was deployed based on whether the type of encephalitis is infectious or autoimmune. The overall effect for each subgroup is reported when at least two studies exist in the subgroup. In addition, the comparison of the subgroups was performed when each subgroup had at least two studies. Meta-regression was also performed based on the mean age of the participants of each study if it was possible (more than two studies with reported age were present in the meta-analysis; studies that did not report the mean or median age of the participants were omitted from meta-regression). Funnel plot asymmetry and Egger's test were used for evaluation of the publication bias in the included studies. To further assess the source of the heterogeneity, sensitivity analysis was used to determine potentially influential cases. Each time one study was removed, and the effect size was recalculated to optimize the robustness of the combined effect estimate and examine its influence on the pooled SMD. All statistical analysis has been performed using the "meta" package of R software (version 4.1.1) [38].

## 3. Results

### 3.1. Study selection

The search was conducted using EMBASE, Scopus, PubMed, and Web of Science Databases for related articles, which yielded a total of 8,685 records. After searching for duplicates and removing them, 3,054 results remained for screening, of which 159 articles remained after title/abstract screening. In the process of full-text revision, 88 articles did not meet the eligibility criteria and were excluded for different reasons, as outlined in Fig 1 based on PRISMA guidelines (_((((xxx))))_)[39–103]. In some of the excluded articles, the study was conducted to compare cytokines levels in other illnesses or different stages of encephalitis, all lacking control groups (_((((xxx))))_)[44, 45, 47, 50, 53, 54, 64–66, 69, 71, 72, 74–76, 80–86, 89, 90, 94, 95, 100–103]. There were articles whose results could not be included in this review because of being in vivo, in vitro, case reports, review articles, and conference abstracts (_((((xxx))))_)[39, 40, 42, 46, 49, 51, 52, 55, 57, 59, 61, 63, 67, 68, 73, 78, 79, 93, 97–99]. In 12 studies, there were neither qualitative reports of cytokine levels nor were there quantitative data of cytokine concentrations; therefore, they did not meet our inclusion criteria [41, 48, 56, 58, 60, 62, 70, 77, 87, 88, 91, 92]. Two studies measured the efficacy and side effects of vaccines [43, 96]. The full-text article could not be found in 21 studies (_((((xxx))))_)[80, 104–123]. Based on the inclusion and exclusion criteria, 71 studies (_((((xxx))))_)[8, 23–30, 121, 124–184] were considered for this study, of which 23 studies had sufficient quantitative data and were included in the meta-analysis and reported in this study (_((((xxx))))_)[25, 28–30, 125–127, 131, 134, 140–142, 144, 147, 149, 152, 156, 175, 178–180, 183, 184]. The remaining 48 studies were not included in the meta-analysis because they mostly reported qualitative data and did not contain sufficient quantitative data. Our study does not report qualitative data and only focuses on the meta-analysis of quantitative data. Although Chen et al. [8] study investigated the concentration of cytokines, the reported data overlapped Zou et al. [184] study, published more recently, so we considered the latter in the meta-analysis. The meta-analysis was performed for 12 cytokines and chemokines, each of which was assessed in at least three comparison studies.

### 3.2. Characteristics and quality of the included studies

The included studies' publication time were between 1996 and 2021. CSF was used in 11 studies for evaluating the concentration of the cytokines [29, 30, 125, 126, 134, 140, 141, 149, 156, 178, 184], plasma/serum was examined for cytokines concentration in 2 studies [28, 142], and both CSF and serum/plasma were investigated in 9 studies [25, 127, 131, 144, 147, 152, 175, 179, 180, 183]. Twelve cytokines and chemokines were included in the CSF meta-analysis, namely IL-2 [149, 156, 175], IL-4 [125, 126, 156, 183], IL-6 [25, 125, 126, 134, 141, 156, 178, 184], IL-8 [29, 30, 125, 126, 140, 156], IL-10 [25, 125, 126, 134, 149, 156, 184], IL-17 [25, 125, 141, 178], TNF-α [134, 144, 156, 175, 184], IFN-γ [125, 126, 149, 156, 183], CCL2 [125, 126, 156], CXCL9 [140, 152, 156], CXCL10 [147, 156, 179, 180], and CXCL13 [25, 156, 179]. Five cytokines, including CXCL10 [127, 147, 179, 180], CXCL13 [25, 179, 180], IL-6 [25, 131, 142], IL-10 [25, 28, 131], and TNF-α [28, 142, 144, 175] were encompassed in the serum/plasma meta-analysis. The concentration of the cytokines were measured using enzyme-linked immunosorbent assay (ELISA) (_((((xxx))))_)[25, 28–30, 131, 134, 141, 142, 147, 149, 152, 175, 178–180, 183, 184], flowcytometry immunoassay [28, 127, 140], and multiplex assay [125, 126, 156]. The total number of the included subjects in the serum meta-analysis is 421, comprised of 241 patients and 180 controls. Concerning CSF meta-analysis, a total of 1062 individuals are available, including 670 patients 392 controls. The summary of the characteristics of the studies is available in the Table 1.

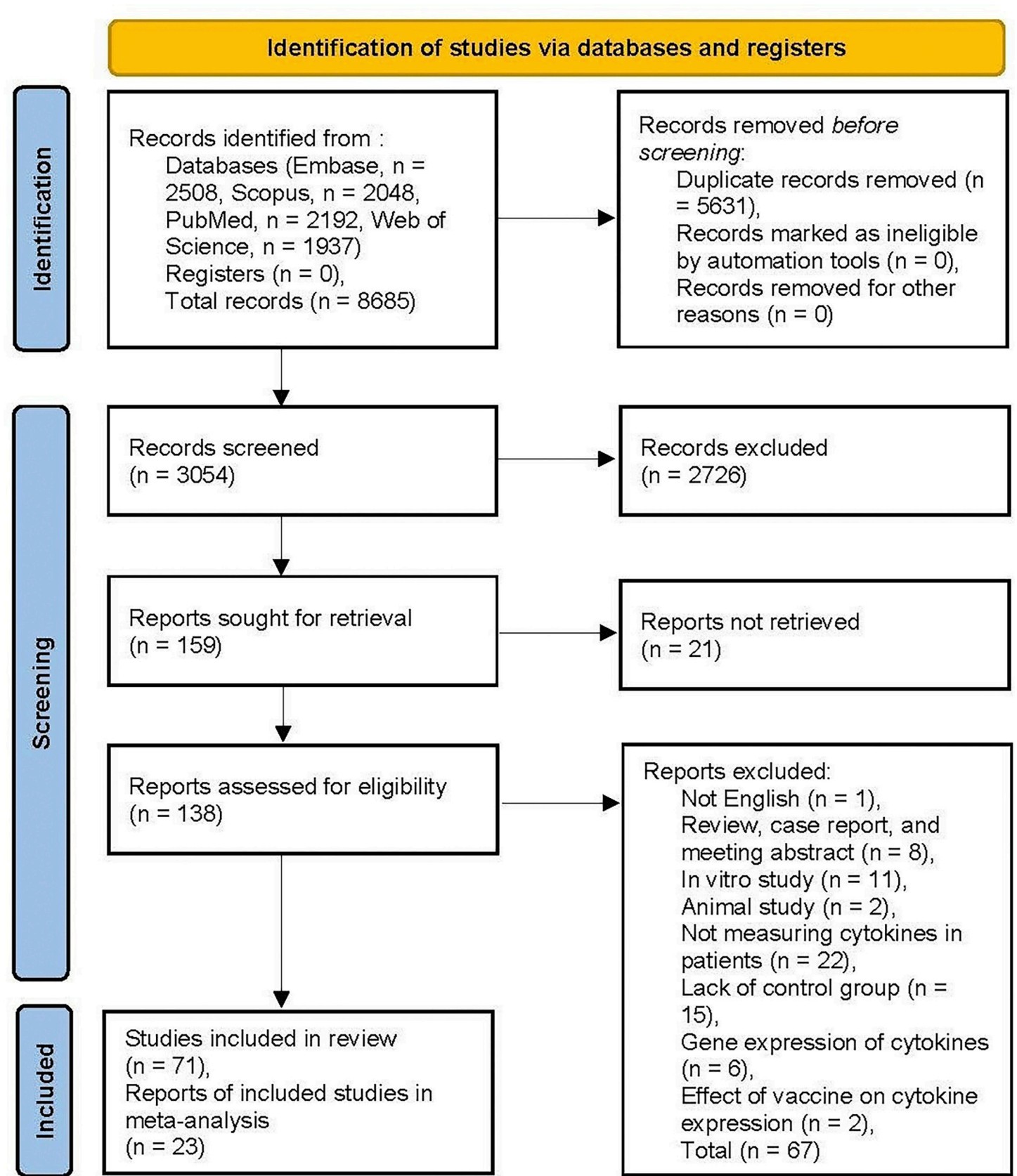

**Fig 1. Flow diagram of literature search and study selection.** The values demonstrate the number of documents in each category.

**Table 1. Summary of findings from studies included in the quantitative analysis of serum/plasma or CSF levels in encephalitis disease.**

| First Author | Year | Country | CSF/ Serum | Measured Cytokine(s) | Encephalitis Type | Patients N | Controls N | Assay Type | Female % | Mean Age | Encephalitis Type |
|---|---|---|---|---|---|---|---|---|---|---|---|
| Tsai, M. L. [30] | 1996 | Taiwan | CSF | IL-8 | Infectious | 11 | 11 | ELISA | - | - | |
| Singh, A. [29] | 2000 | India | CSF | IL-8 | Infectious | 30 | 27 | ELISA | - | - | JE |
| Lin, T. Y. [142] | 2002 | Taiwan | Serum | IL-6, TNF-α | Infectious | 8 | 21 | ELISA | 14.19 | 19.59 | |
| Wang, S. M. [28] | 2003 | Taiwan | Serum | IL-2, IL-4, IL-10, TNF-α, IFN-γ | Infectious | 34 | 15 | Multiplex Assay | - | - | EV71-associated BE |
| Leake, J. A. [149] | 2004 | USA | CSF | IL-2, IL-10, IFN-γ | Auto-Immune | 11 | 28 | ELISA | - | - | ADEM |
| Babu, G. N. [175] | 2006 | India | CSF & Serum | IL-2, TNF-α | Infectious | 18 | 20 | ELISA | - | - | JE |
| Lepej, S. Z. [147] | 2007 | Croatia | CSF & Serum | CXCL10 | Infectious | 19 | 10 | Immunoassay | 50.34 | 34.90 | TBE patient |
| Wang, S. M. [127] | 2008 | Taiwan | CSF & Serum | CCL2, IL-8, CXCL10 | Infectious | 21 | 13 | Multiplex Assay | - | - | EV71-associated BE |
| Zajkowska, J. [179] | 2011 | Poland | CSF & Serum | CXCL10, CXCL13 | Infectious | 15 | 8 | ELISA | 33.30 | 43.00 | TBE |
| Ygberg, S. [125] | 2016 | Sweden | CSF | IL-4, IL-6, IL-8, IL-10, IL-17, CCL2, IFN-γ | Both | 17 | 13 | Multiplex Assay | 64.68 | 51.29 | 13 anti-NMDAR + 4 Infectious |
| Singh, S. K. [131] | 2017 | India | CSF & Serum | IL-6, IL-10 | Both | 87 | 64 | ELISA | 98.00 | - | 13 AES+JE, 74 AES |
| Koper, O. M. [152] | 2018 | Poland | CSF & Serum | CXCL9 | Infectious | 24 | 13 | ELISA | 64.24 | 47.11 | TBE |
| Ai, P [178] | 2018 | China | CSF | IL-6, IL-17 | Auto-Immune | 33 | 38 | ELISA | 54.46 | 37.55 | Anti-NMDAR |
| Maric, L. S. [180] | 2018 | Croatia | CSF & Serum | CXCL10, CXCL13 | Auto-Immune | 23 | 20 | ELISA | - | - | ADEM |
| Liu, B. [141] | 2018 | China | CSF | IL-6 | Auto-Immune | 24 | 31 | ELISA | 47.28 | 36.98 | NMDAR |
| Liu, J. [140] | 2018 | China | CSF | IL-8, IL-10, CXCL9 | Infectious | 99 | 22 | Multiplex Assay | 39.67 | 28.79 | EV71-associated BE |
| Lin, Y. T. [25] | 2019 | China | CSF & Serum | IL-6, IL-10, IL-17, CXCL13 | Auto-Immune | 16 | 9 | ELISA | 15.92 | 51.65 | Anti-LGI1 encephalitis |
| Peng, Y. [134] | 2019 | China | CSF | IL-10, TNF-α | Auto-Immune | 33 | 17 | ELISA | 56.00 | 34.87 | NMDAR |
| Jiang, X. Y. [156] | 2020 | China | CSF | IL-2, IL-4, IL-6, IL-8, IL-10, TNF-α, CCL2, CXCL9, CXCL10, CXCL13, IFN-γ | Auto-Immune | 147 | 35 | Multiplex Assay | - | - | NMDAR |
| Ygberg, S. [126] | 2020 | Sweden | CSF | IL-4, IL-6, IL-8, IL-10, CCL2, IFN-γ | Infectious | 37 | 19 | Multiplex Assay | 41.04 | 95.93 | TBE |
| Zou, C. [184] | 2020 | China | CSF | IL-6, TNF-α | Both | 46 | 21 | - | 35.70 | 61.54 | 33 NMDAR + 13 Viral |
| Li, J. [144] | 2020 | China | CSF & Serum | TNF-α | Infectious | 84 | 50 | ELISA | 44.06 | 6.37 | TBE |
| Xie, J. [183] | 2021 | China | CSF & Serum | IL-4, IFN-γ | Infectious | 80 | 40 | ELISA | 52.50 | 7.83 | 40 Viral + 40 Supporative |

Abbreviations: CSF, cerebrospinal fluid; IL-8, interleukin-8; ELISA, enzyme-linked immunoassay; JE, Japanese encephalitis; TNF-a, tumor necrosis factor-a; EV-71, enterovirus-71; BE, brainstem encephalitis; ADEM, acute disseminated *encephalomyelitis*; TBE, tick-borne encephalitis; NMDAR, N-methyl-D-aspartate receptor; AES, acute encephalitis syndrome; LGI1, leucine-rich glioma inactivated1; IFN-γ, Interferon γ

**Table 2. Assessment of the quality of the included studies using Joanna Briggs Institute's (JBI) checklist.**

| Study | Criteria and corresponding scores | | | | | | | | Total | % |
|---|---|---|---|---|---|---|---|---|---|---|
| | #1 | #2 | #3 | #4 | #5 | #6 | #7 | #8 | | |
| Tsai, M. L. 1996 [30] | 0 | 0 | 0 | 1 | NR | NR | 1 | 1 | 3 | 37.5 |
| Singh, A. 2000 [29] | 1 | 1 | 1 | 1 | NR | NR | 1 | 1 | 6 | 75 |
| Lin, T. Y. 2002 [142] | 1 | 1 | 0 | 0 | 1 | 1 | 1 | 1 | 6 | 75 |
| Wang, S. M. 2003 [28] | 1 | 1 | 1 | 1 | NR | NR | 1 | 1 | 6 | 75 |
| Leake, J. A. 2004 [149] | 1 | 1 | 1 | 1 | NR | NR | 1 | 1 | 6 | 75 |
| Babu, G. N. 2006 [175] | 1 | 1 | 0 | 0 | NR | NR | 1 | 1 | 4 | 50 |
| Lepej, S. Z. 2007 [147] | 1 | 1 | 1 | 1 | NR | NR | 1 | 1 | 6 | 75 |
| Wang, S. M. 2008 [127] | 1 | 1 | 1 | 1 | NR | NR | 1 | 1 | 6 | 75 |
| Zajkowska, J. 2011 [179] | 1 | 1 | 1 | 1 | NR | NR | 1 | 1 | 6 | 75 |
| Ygberg, S. 2016 [125] | 1 | 1 | 1 | 1 | NR | NR | 1 | 1 | 6 | 75 |
| Singh, S. K. 2017 [131] | 1 | 1 | 1 | 0 | NR | NR | 1 | 1 | 5 | 62.5 |
| Koper, O. M. 2018 [152] | 1 | 1 | 1 | 1 | NR | NR | 1 | 1 | 6 | 75 |
| Ai, P 2018 [178] | 1 | 1 | 0 | 1 | NR | NR | 1 | 1 | 5 | 62.5 |
| Maric, L. S. 2018 [180] | 1 | 1 | 1 | 1 | NR | NR | 1 | 1 | 6 | 75 |
| Liu, B. 2018 [141] | 1 | 1 | 1 | 1 | NR | NR | 1 | 1 | 6 | 75 |
| Liu, J. 2018 [140] | 1 | 1 | 0 | 1 | NR | NR | 1 | 1 | 5 | 62.5 |
| Lin, Y. T. 2019 [25] | 1 | 1 | 1 | 1 | NR | NR | 1 | 1 | 6 | 75 |
| Peng, Y. 2019 [134] | 1 | 1 | 1 | 1 | 1 | 1 | 1 | 1 | 8 | 100 |
| Jiang, X. Y. 2020 [156] | 1 | 1 | 0 | 1 | NR | NR | 1 | 1 | 5 | 62.5 |
| Li, J. 2020 [144] | 1 | 1 | 0 | 1 | 1 | 1 | 1 | 1 | 7 | 87.5 |
| Ygberg, S. 2020 [126] | 1 | 1 | 1 | 1 | NR | NR | 1 | 1 | 6 | 75 |
| Zou, C. 2020 [184] | 1 | 1 | 1 | 1 | NR | NR | 1 | 1 | 6 | 75 |
| Xie, J. 2021 [183] | 1 | 1 | 0 | 1 | NR | NR | 0 | 1 | 4 | 50 |

#1: Were the criteria for inclusion in the sample clearly defined? #2: Were the study subjects and the setting described in detail? #3: Was the exposure measured in a valid and reliable way? #4: Were objective, standard criteria used for measurement of the condition? #5: Were confounding factors identified? #6: Were strategies to deal with confounding factors stated? #7: Were the outcomes measured in a valid and reliable way? #8: Was appropriate statistical analysis used?

Abbreviations: NR, not reported

0: The criterion is not fulfilled or reported by the study

1: The criterion is met by the study

Using the JBI quality assessment tool, most of the studies achieved a quality score of 5 (62.5) or more, two studies had a score of 4 (50%) [175, 183], and the score of one study was 3 (37.5) [30]. A summary of the quality assessment of included studies is provided in Table 2. The funnel, drapery, and sensitivity analysis forest plots of the analyses are available in S1 File.

### 3.3. IL-2

Three studies [149, 156, 175] with 176 encephalitis patients and 83 controls reported CSF concentration of IL-2. Although the overall concentration of the IL-2 was higher in encephalitis patients, this difference was not significant, with a $P$-value of 0.05 (SMD, 0.82; 95% CI, -0.02–1.66). The heterogeneity was considerable with $Q = 11.58$ ($P < 0.01$) and $I^2 = 83\%$. The meta-regression for age was not applicable because data for age were only available in one study. In addition, subgroup analysis demonstrated non-significant increase in the concentration of IL-2 in infectious encephalitis (SMD = 0.87; 95% CI, -0.89–2.63; $P = 0.33$) with high heterogeneity ($Q = 11.57$; $P < 0.01$; $I^2 = 91\%$). There was only one study available for the autoimmune subgroup; hence, the overall effect for this subgroup and comparison between subgroups are not applicable. The

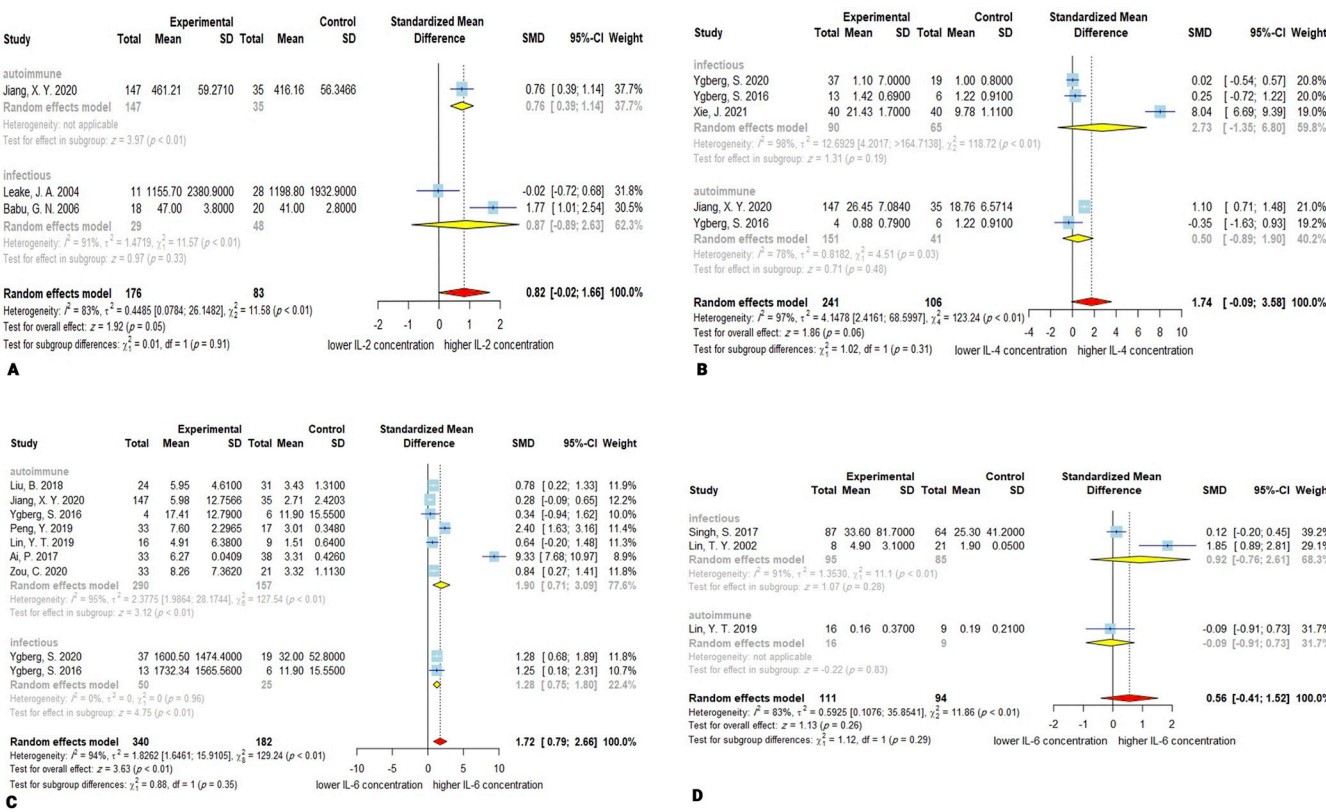

**Fig 2. Forest plots for results of meta-analysis for IL-2 CSF concentrations (A), IL-4 CSF concentrations (B), IL-6 CSF concentrations (C), and IL-6 serum/plasma concentrations (D).** SD, standard deviation; SMD, standardized mean difference; CI, confidence interval.

Egger's test revealed no publication bias ($P = 0.91$). Sensitivity analysis revealed that omitting Leake, J. A. 2004 study leads the overall effect to become significant ($P = 0.02$) (Fig 2A).

### 3.4. IL-4

241 patients and 106 controls from 4 studies [125, 126, 156, 183] were included in the meta-analysis of CSF concentration of IL-4. The meta-analysis showed slightly elevated IL-4 CSF levels, but this was not statistically significant (SMD, 1.74; 95% CI, -0.09–3.58; $P = 0.06$). The heterogeneity was significant with $Q = 123.24$ ($P < 0.01$) and $I^2 = 97\%$. The meta-regression demonstrated no significant effect for age as a moderator of the meta-analysis ($P = 0.33$). The subgroup analysis did not show significant alteration in IL-4 levels in both infectious (SMD, 2.73; 95% CI, -1.35–6.80; $P = 0.19$) or autoimmune encephalitis (SMD, 0.50; 95% CI, -0.89–1.90; $P = 0.48$). The autoimmune subgroup had less heterogeneity ($Q = 4.51$; $P < 0.05$; $I^2 = 78\%$) than the infectious group ($Q = 118.72$; $P < 0.01$; $I^2 = 98\%$). The chi-square test for assessing the disparity between subgroups demonstrated no significant difference ($P = 0.31$). There was no publication bias based on the Egger's linear regression test for funnel plot asymmetry ($P = 0.56$). Omitting Ygberg, S. 2016 from autoimmune subgroup in the sensitivity analysis resulted in a significant overall effect ($P = 0.04$) (Fig 2B).

### 3.5. IL-6

Out of 10 studies that investigated the concentration of the IL-6, 7 reported its concentration in CSF [125, 126, 134, 141, 156, 178, 184], 2 reported its concentration in serum/plasma [131, 142], and 1 study reported both serum/plasma and CSF levels of IL-6 [25].

**3.5.1. CSF IL-6.** A total of 340 patients and 182 controls explored CSF concentration of IL-6. The CSF levels of the IL-6 in encephalitis patients are significantly higher than controls (SMD, 1.72; 95% CI, 0.79–2.66; $P < 0.001$). The overall heterogeneity is high, with $Q = 129.24$ ($P < 0.01$) and $I^2 = 94\%$. Meta-regression for age as moderator was not significant ($P = 0.34$). Subgroup analysis results are consistent and show that in both autoimmune and infectious encephalitis, CSF concentration of IL-6 is higher than controls (SMD (autoimmune), 1.90; 95% CI, 0.71–3.90; $P < 0.01$; SMD (infectious), 1.28; 95% CI, 0.75–1.80; $P < 0.01$). There was no significant difference between the two subgroups ($P = 0.35$). The tests for heterogeneity showed a high heterogeneity in the autoimmune subgroup ($Q = 127.54$; $P < 0.01$; $I^2 = 95\%$) and a lack of observed heterogeneity in the infectious subgroup ($Q = 0.00$; $P = 0.96$; $I^2 = 0.0\%$). The publication bias was not significant based on the Egger's test ($P = 0.05$), and the sensitivity analysis demonstrated that Ai, P. can influence the overall effect size, without changing the statistical significance (Fig 2C).

**3.5.2. Serum IL-6.** Pooled data demonstrated that the serum/plasma levels of the IL-6 are not significantly higher (SMD, 0.56; 95% CI, -0.41–1.52; $P = 0.26$) in encephalitis patients ($n = 111$) than controls (n = 94). The overall heterogeneity was high ($Q = 11.86$; $P < 0.01$; $I^2 = 83\%$). Meta-regression was not applicable due to the lack of enough data. Subgroup analysis for infectious encephalitis indicated similar results to the overall pooled effect with high heterogeneity (SMD, 0.92; $P = 0.28$; 95% CI, -0.76–2.61; $Q = 11.1$, $P < 0.01$; $I^2 = 91\%$). Since there is only one study in the autoimmune subgroup, the overall effect for this subgroup and comparison between subgroups are not applicable. The Egger's test showed no significant publication bias ($P = 0.58$). The sensitivity analysis suggested that omitting Singh, S. 2017 or Lin, Y. T. 2019 could affect the overall effect to be statistically significant. This happens because of the small number of studies and high SMD of the Lin, T. Y. 2002 study, which could affect the pooled effect in the absence of the mentioned studies (Fig 2D).

## 3.6. IL-8

Random effect meta-analysis of 6 studies [29, 30, 125, 126, 140, 156] demonstrated that encephalitis patients ($n = 341$) had higher CSF IL-8 concentration compared to control group ($n = 126$) for IL-8 (SMD, 1.03; 95% CI, 0.17–1.90; $P < 0.05$). The overall heterogeneity calculated is $Q = 70.63$ ($P < 0.01$), $I^2 = 92\%$. Meta-regression demonstrated significant effect of age on the CSF concentration of IL-8 ($P < 0.05$). In subgroup analysis, the CSF concentration of IL-8 in the autoimmune subgroup was significantly higher than controls (SMD, 1.15; 95% CI, 0.77–1.52; $P < 0.01$). In infectious subgroup, IL-8 CSF levels were not statistically different from controls (SMD, 1.09; 95% CI, -0.19–2.38; $P = 0.10$). The most observed heterogeneity was attributable to the infectious subgroup ($Q = 66$; $P < 0.01$; $I^2 = 94\%$), and there was no considerable observed heterogeneity in autoimmune subgroup ($Q = 0.79$; $P = 0.37$; $I^2 = 0.0\%$). The Between-subgroup test did not show any significant difference ($P = 0.94$). The Egger's linear regression test for meta bias reported that publication bias was not significant ($P = 0.68$). Sensitivity analysis indicates that removing Singh, A. 2000 and Jiang, X. Y. 2020 studies leads to a non-significant overall effect. This is possibly due to the high effect size reported in Singh, A. 2000 study and robust large sample size in Jiang, X. Y. 2020, which greatly influences the overall result (Fig 3A).

## 3.7. IL-10

Nine studies [25, 28, 125, 126, 131, 134, 149, 156, 184] have investigated the concentration of IL-10: 6 studies in CSF [125, 126, 134, 149, 156, 184], 2 studies in serum/plasma [28, 131], and 1 study in both [25].

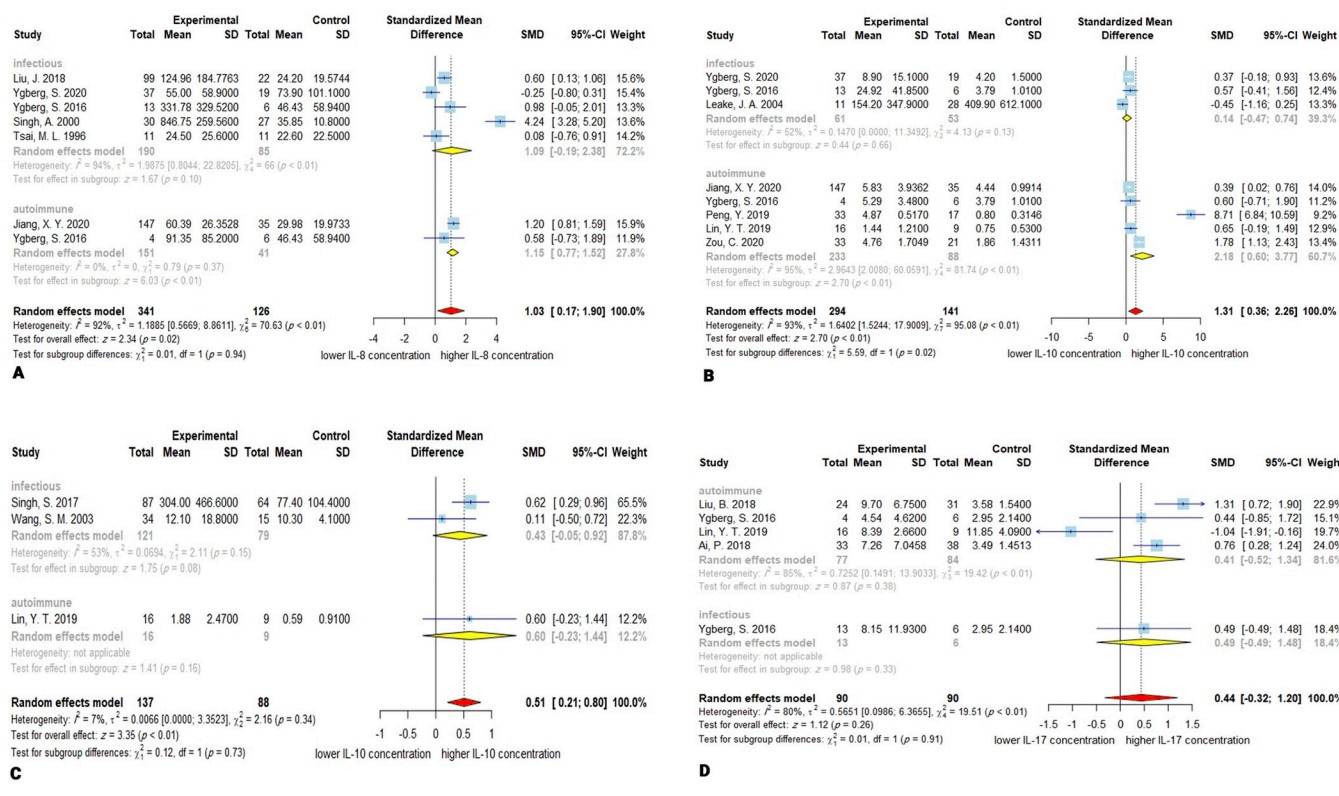

**Fig 3.** Forest plots for results of meta-analysis for IL-8 CSF concentrations **(A)**, IL-10 CSF concentration **(B)**, IL-10 serum/plasma concentrations **(C)**, and IL-17 CSF concentrations **(D)**. SD, standard deviation; SMD, standardized mean difference; CI, confidence interval.

### 3.8. CSF IL-10

The higher CSF concentration of IL-10 in encephalitis patients ($n = 294$) compared to controls ($n = 141$) was confirmed by meta-analysis (SMD, 1.31; 95% CI, 0.36–2.26; $P < 0.01$). The heterogeneity among the studies is significant ($Q = 95.08$; $P < 0.0001$; $I^2 = 93\%$). The meta-regression demonstrated no significant effect of the age on the CSF concentration of the IL-10 ($P = 0.24$). In the autoimmune subgroup the effect was consistent with the overall effect (SMD, 2.18; 95% CI, 0.60–3.77; $P < 0.01$), but the infectious subgroup exhibited no significant alteration of the CSF IL-10 levels (SMD, 014; 95% CI, -0.47–0.74; $P = 0.66$). The test for the difference between subgroups was significant ($P = 0.02$). The existing heterogeneity can be mostly ascribed to the autoimmune subgroup with $Q = 81.74$ ($P < 0.01$; $I^2 = 95$), and the heterogeneity in the infectious was moderate ($Q = 4.13$; $P = 0.13$; $I^2 = 52\%$). The difference between subgroups was significant with $P$-value of 0.02. The Egger's test for the publication bias was not significant ($P = 0.16$). The sensitivity analysis revealed no single study with significant impact of the overall effect (Fig 3B).

### 3.9. Serum IL-10

Similarly, the concentration of the IL-10 in serum/plasma was higher in encephalitis patients ($n = 137$) than controls ($n = 88$; SMD, 0.51; 95% CI, 0.21–0.80; $P < 0.001$). The between studies heterogeneity was low ($Q = 2.16$; $P = 0.34$; $I^2 = 7\%$). The meta-regression for age was not applicable since the data for age was only available in one study. There was only one study in the autoimmune subgroup, and the pooled effect and evaluation of the heterogeneity were not

applicable. Serum/plasma alteration of IL-10 levels in the infectious group was not significant (SMD, 0.43; 95% CI, -0.05–0.92; $P$ = 0.08). The heterogeneity was moderate ($Q$ = 2.11; $P$ = 0.15; $I^2$ = 53%). The publication bias was not significant ($P$-value = 0.66). Furthermore, the sensitivity analysis demonstrated that omitting either Singh, S. 2017, or Lin, Y. T. 2019 will make the overall effect non-significant, which may be a consequence of the low number of studies available for the meta-analysis (Fig 3C).

### 3.10. IL-17

Four studies reported the CSF concentration of IL-17 or IL-17A for encephalitis patients ($n$ = 90) and control group ($n$ = 90). The overall levels of IL-17 were not significantly higher in encephalitis patients compared with controls (SMD, 0.44; 95% CI, -0.32–1.20; $P$ = 0.26). The heterogeneity was high with $Q$ = 19.51 ($P$ < 0.001), and $I^2$ = 80%. The meta-regression demonstrated a non-significant effect of age on the CSF concentration of the IL-17 ($P$ = 0.54). The subgroup analysis demonstrated that in autoimmune subgroup IL-17 levels were not significantly altered (SMD, 0.41; 95% CI, -0.52–1.34; $P$ = 0.38), with a high heterogeneity among studies ($Q$ = 19.42; $P$ < 0.01; $I^2$ = 85%). The pooled estimates were not performable for the infectious subgroup because only one study belongs to it. The Egger's test for funnel plot asymmetry reported no significant publication bias ($P$ = 0.40). The sensitivity analysis demonstrated that removing Lin, Y. T. 2019, because of its reverse reported effect, increases the overall estimate and leads to significantly higher IL-17 levels in patients than controls ($P$ < 0001) (Fig 3D).

### 3.11. CCL2

Three studies with 201 encephalitis patients and 66 controls investigated the CSF concentration of CCL2. The overall effect tended to show decreased concentration of CCL2 (SMD, -0.53; 95% CI, -1.08–0.01; $P$ = 0.06). The between-study heterogeneity was moderate ($Q$ = 7.11; $P$ = 0.07; $I^2$ = 58%). Moreover, the meta-regression demonstrated no significant moderating effect for age in the meta-analysis ($P$ = 0.10). In the autoimmune subgroup the overall CCL2 levels were significantly decreased (SMD, -0.54; 95% CI, -0.90–-0.18; $P$ <0.01), but this reduction was not significant in the infectious subgroup (SMD, -0.40; 95% CI, -1.89–1.10; $P$ = 0.60). There was no difference between subgroups ($P$ = 0.86). There was no heterogeneity in the autoimmune subgroup ($Q$ = 0; $P$ = 1; $I^2$ = 0%), and the infectious subgroup is mostly responsible for the overall heterogeneity ($Q$ = 6.85; $P$ < 0.01; $I^2$ = 85%). Publication bias was not significant ($P$ = 0.70). Based on the sensitivity analysis, omitting the Ygberg, S. 2016 from infectious subgroup will result in significant overall effect size (Fig 4A).

### 3.12. CXCL9

Three studies with 270 encephalitis subjects and 70 control subjects reported the CSF concentration of the CXCL9. The overall effect shows the CSF concentration of the CXCL9 is higher in encephalitis than in control, whereas non-significant (SMD, 2.50; 95% CI, -0.15–5.16; $P$ = 0.06). The heterogeneity is substantially high ($Q$ = 99.75; $P$ < 0.01; $I^2$ = 98%). The meta-regression for age was not applicable since the data for age was only available in two studies. The subgroup analysis for the infectious encephalitis revealed significantly increased levels of CXCL9 in the CSF (SMD, 3.58; 95% CI, 3.02–4.14; $P$ < 0.01). There was no observed heterogeneity in the infectious subgroup ($Q$ = 0.56; $P$ = 0.46; $I^2$ = 0%). Egger's linear regression for the funnel plot asymmetry exhibited no significant publication bias ($P$ = 0.33). The estimates for the autoimmune subgroup were not applicable because there is only one study. The removal of the Jiang, X. Y. 2020 effect in the sensitivity study causes the overall effect to be significant ($P$ < 0.0001) (Fig 4B).

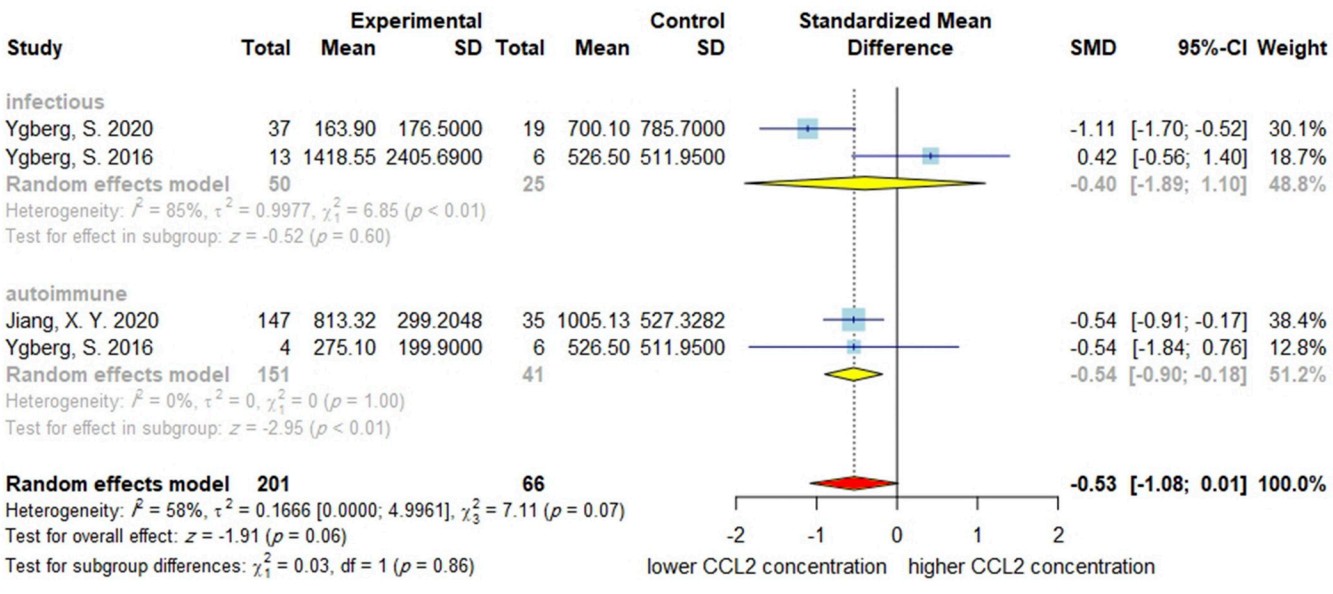

**A**

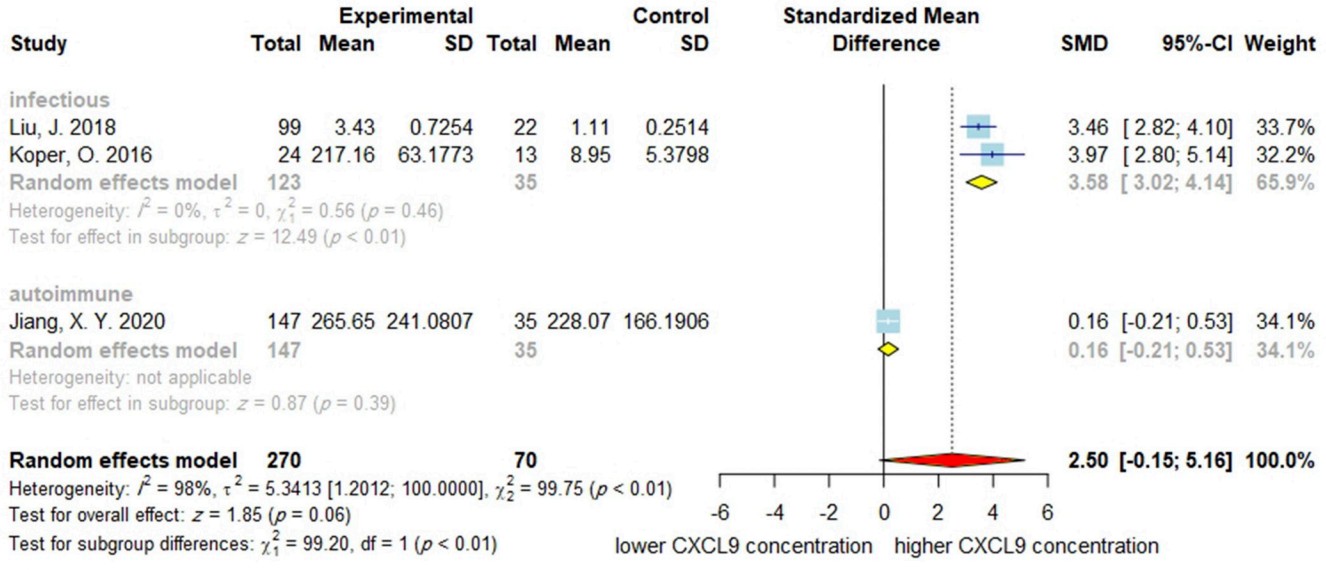

**B**

**Fig 4.** Forest plots for results of the meta-analysis for CCL2 CSF concentrations (**A**) and CXCL9 CSF concentrations (**B**). SD, standard deviation; SMD, standardized mean difference; CI, confidence interval.

### 3.13. CXCL10

Out of 5 studies that reported the concentration of the CXCL10, one measured CSF concentration of CXCL10 [156], one measured serum/plasma [127], and three measured both concentrations [147, 179, 180].

**3.13.1. CSF CXCL10.** Meta-analysis of 204 encephalitis patients and 73 controls showed that the CSF levels of the CXCL10 are higher in the encephalitis group than in the control

group (SMD, 0.86; 95% CI, 0.16–1.56; $P = 0.02$). The overall heterogeneity among studies is high ($Q = 13.25$; $P < 0.01$; $I^2 = 77\%$). The meta-regression for age as a moderator was not significant ($P = 0.39$). The subgroup analysis in the infectious encephalitis, similar to the overall meta-analysis, demonstrated a higher CSF concentration of the CXCL10 in the encephalitis patients compared to the controls (SMD, 1.16; 95% CI, 0.22–2.10; $P = 0.02$). The heterogeneity in infectious subgroup was moderate ($Q = 7.72$; $P = 0.02$; $I^2 = 74\%$). The autoimmune subgroup only had one study, and overall estimates were not applicable. Publication bias based on Egger's linear regression test was not significant ($P = 0.08$). Sensitivity analysis revealed that omitting the Maric, L. S. 2018 study will cause the overall effect to become non-significant (Fig 5A).

**3.13.2. Serum CXCL10.** In contrast to the CSF meta-analysis, Serum analysis reported no significant alteration in the CXCL10 concentration ($n_{patients} = 78$; $n_{controls} = 51$; SMD, 0.12; 95% CI, -1.27–1.51; $P = 0.87$). The observed heterogeneity was high with $Q = 37.08$ ($P < 0.0001$; $I^2 = 92\%$). The meta-regression for age as a moderator variable was not significant ($P = 0.52$). The subgroup analysis was not conductible since the included studies only belong to infectious encephalitis. The publication bias was not significant ($P = 0.53$). The sensitivity analysis revealed that omitting the studies could lead to a change in the direction of the overall effect. Except for Lepej, S. Z. 2007, whose omission leads to a slightly significant overall effect, removal of the other studies does not lead to a significant change in the overall effect (Fig 5B).

## 3.14. CXCL13

Among 4 studies that measured CXCL13 concentration, one investigated the CSF concentration [156], one assessed serum/plasma concentration [180], and two measured both CSF and serum/plasma concentration [25, 179].

**3.14.1. CSF CXCL13.** 178 encephalitis patients and 52 controls entered the CSF meta-analysis for CXCL13. There was a non-statistically significant increase in the concentration of CXCL13 in patients (SMD, 091; 95% CI, -0.23–2.05; $P = 0.12$). The overall heterogeneity was high with $Q = 13.57$ ($P < 0.01$; $I^2 = 85\%$). The subgroup analysis in autoimmune encephalitis group demonstrated no significant alteration of the CXCL13 concentration with high between-study heterogeneity (SMD, 1.39; 95% CI, -0.66–3.44; $P = 0.18$; $Q = 12.23$; $P < 0.01$; $I^2 = 92\%$). The meta-regression for age was not applicable since the data for age was only available in two studies. The estimates for the infectious subgroup are not available since there was only one study. Publication bias based on Egger's test was not significant ($P = 0.59$). The sensitivity analysis revealed that the removal of Lin, Y. T. 2019 could lead to a significant overall effect size with $P$-value = 0.04 (Fig 5C).

**3.14.2. Serum CXCL13.** 54 cases and 37 controls were investigated for serum/plasma concentration of the CXCL13. The meta-analysis came up with no significant difference in the serum concentration of CXCL13 in encephalitis patients and controls (SMD, 0.04; 95% CI, -1.48–1.56; $P = 0.96$). The sample size was small, and CI was wide. The heterogeneity was high with $Q = 21.38$ ($P < 0.0001$; $I^2 = 91\%$). The meta-regression demonstrated a significant effect of age on the serum concentration of CXCL13 ($P < 0.0001$). The subgroup analysis results are only available for the infectious encephalitis group, which, similar to the overall effect, does not implicate any significant alteration in CXCL13 concentration (SMD, -0.52; 95% CI, -2.13–1.10; $P = 0.53$). The heterogeneity in infectious subgroup was also high ($Q = 8.74$; $P < 0.01$; $I^2 = 89\%$). There was no significant publication bias ($P = 0.13$). Sensitivity analysis reported that omitting Lin, Y. T. 2019 study leads to a slightly significant overall effect size ($P = 0.04$) (Fig 5D).

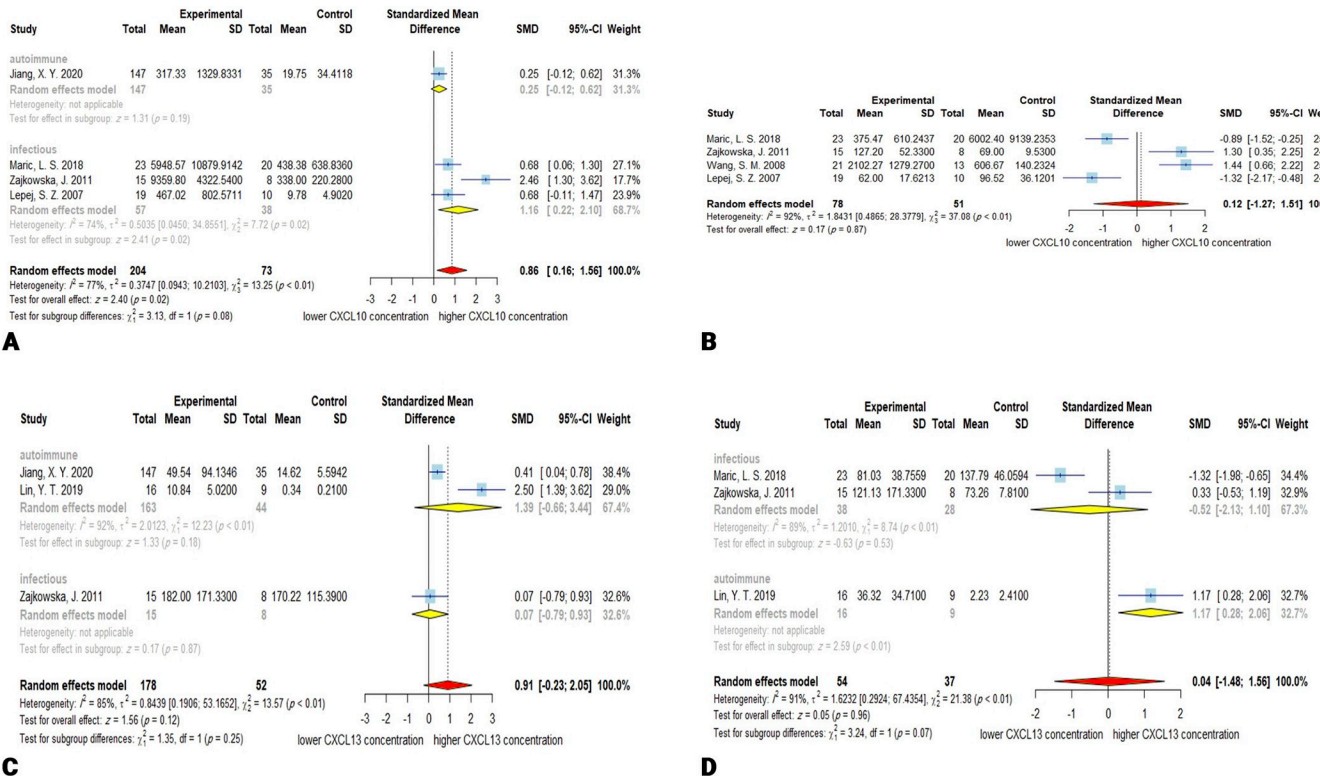

**Fig 5.** Forest plots for results of the meta-analysis for CXCL10 CSF concentrations (**A**), CXCL10 serum/plasma concentrations (**B**), CXCL13 CSF concentrations (**C**), and CXCL13 serum/plasma concentrations (**D**). SD, standard deviation; SMD, standardized mean difference; CI, confidence interval.

### 3.15. TNF-α

Seven studies investigated the concentration of TNF-α: two were related to serum/plasma concentration [28, 142], three to CSF concentration [134, 156, 184], and two measured both CSF and serum/plasma concentration [144, 175].

**3.15.1. CSF TNF-α.** Increased CSF concentration of the TNF-α was seen in the meta-analysis of 213 encephalitis patients and 73 controls (SMD, 2.61; 95% CI, 1.29–3.94; $P < 0.001$). The observed heterogeneity was high with $Q = 90.82$ ($P < 0.0001$; $I^2 = 96\%$). The meta-regression demonstrated a significant effect of age on the CSF concentration of CXCL13 ($P < 0.0001$). The effect in the autoimmune encephalitis sub-group was similar to the overall effect (SMD, 1.77; 95% CI, 0.45–3.09; $P < 0.01$). Subgroup-analysis for infectious encephalitis demonstrated no significant alteration in the concentration of the TNF-α (SMD, 10.91; 95% CI, -9.02–30.84; $P = 0.28$). The test for subgroup difference demonstrated no significant difference ($P = 0.37$). The heterogeneity was high in both infectious ($Q = 61.13$; $P < 0.01$; $I^2 = 98\%$) and autoimmune ($Q = 29.21$; $P < 0.01$; $I^2 = 93\%$) subgroups. The Egger's linear regression test for funnel plot asymmetry demonstrated significant publication bias with $P$-value less than 0.001. The sensitivity analysis came up with no change in the significance of the effect when putting each of studies aside (Fig 6A).

**3.15.2. Serum TNF-α.** The meta-analysis demonstrated that the concentration of the TNF-α is increased in the encephalitis patients compared to controls ($n_{patients} = 144$; $n_{control} = 106$; SMD, 9.06; 95% CI, 2.85–15.27; $P < 0.01$). The observed heterogeneity of the reported studies was high ($Q = 317.69$; $P < 0.0001$; $I^2 = 99\%$). The meta-regression for age was not applicable since the data for was only available in two studies. The subgroup analysis was not

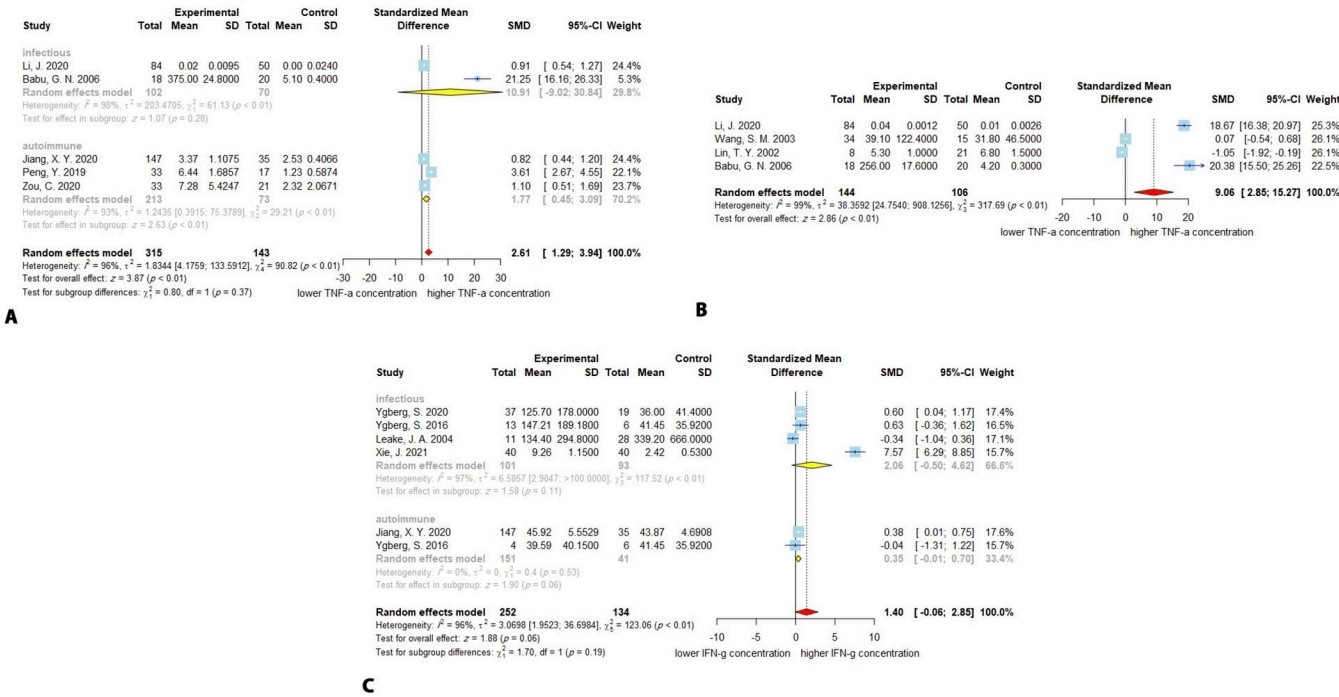

**Fig 6.** Forest plots for results of the meta-analysis for TNF-α CSF concentrations (**A**), TNF-α serum/plasma concentrations (**B**), and IFN-γ CSF concentrations (**C**). SD, standard deviation; SMD, standardized mean difference; CI, confidence interval.

available because all studies investigated infectious encephalitis patients. Publication bias was not significant with *P*-value = 0.14. The sensitivity analysis displayed that omitting each study except Li, J. 2020, which does not change the significance of the overall effect, will lead to a non-significant overall effect size (Fig 6B).

## 3.16. IFN-γ

252 encephalitis patients and 134 controls were included in the IFN-γ CSF concentration meta-analysis, which showed a tendency to increased concentration of IFN-γ in the former group (SMD, 1.40; 95% CI, -0.06–2.85; *P* = 0.06). The overall heterogeneity was high ($Q$ = 123.06; *P* < 0.0001; $I^2$ = 96%). The meta-regression demonstrated no significant effect of the age on the CSF concentration of the IFN-γ (*P* = 0.30). The effects in the infectious (SMD, 2.06; 95% CI, -0.50–4.62; *P* = 0.11) and autoimmune (SMD, 0.35; 95% CI, -0.01–0.70; *P* = 0.06) subgroups were consistent with the overall effect. The heterogeneity in infectious subgroups was high ($Q$ = 117.52; *P* < 0.01; $I^2$ = 97%), but the heterogeneity in autoimmune subgroup was low ($Q$ = 0.4; *P* = 0.53; $I^2$ = 0%). Publication bias was not significant with *P*-value = 0.34. Sensitivity analysis via leaving one study out at each time, showed that omitting Ygberg, S. 2016 and Leake, J. A. 2004 can lead to increased overall effect with a *P*-value of 0.046 and 0.050, respectively (Fig 6C).

## 4. Discussion

The present meta-analysis was conducted to explore cytokine levels in patients with encephalitis compared to controls without encephalitis, as well as to compare the cytokine concentrations in autoimmune encephalitis and infectious encephalitis. The cytokines IL-6, IL-8, IL-10, CXCL10, and TNF-α were significantly higher in the CSF of patients compared to controls.

Also, there was a significant difference between infectious and autoimmune encephalitis regarding CSF levels of IL-10 and CXCL9. Increased serum levels of TNF-α were observed in patients with encephalitis. The Meta-analyses results have been summarized in Table 3, Figs 7 and 8, S1 and S2 Figs.

Encephalitis is a condition that can be the result of various causes. It can be classified into infectious and autoimmune encephalitis. Infectious encephalitis is caused when an infectious agent enters the body, mainly viruses, but some cases of primary encephalitis are caused by bacteria, fungi, or parasites. The common viral causes of encephalitis include HSV, JEV, varicella zoster, measles, mumps, rabies, and rubella [185–187].

Herpes simplex encephalitis (HSE) is one of the most common types of infectious encephalitis. It is caused by the HSV-1 virus–a neurotropic virus that infects the trigeminal nerve of humans [188]. Despite its widespread prevalence, it is still unknown whether HSE is caused by viral reactivation inside the trigeminal ganglia, direct primary infection of the olfactory mucosa, or other infected central nervous system (CNS) neurons in the brain [189]. Prior investigations have demonstrated that cytokines, such as TNF-α, IFN-γ, and IL-1, play an important role in the pathogenesis of HSE [190, 191].

When the body is infected with HSV, toll-like receptors (TLRs) such as TLR2, TLR3, TLR7, and TLR9 help identify the virus and then, through signaling cascades, trigger intracellular responses. Consequently, these responses stimulate the activation and migration of immune cells [191, 192]. Moreover, the activation of TLRs via HSV-1 proteins or nucleic acids can result in the expression of chemokines and pro-inflammatory cytokines that accelerate the activation of innate immune reactions [93, 193, 194]. The immune reactions stimulate numerous types of immune cells to produce a wide range of cytokines and chemokines (CXCL9), including interleukins, TNF-α, and interferon regulatory factors (IRF3, IRF7, and type I IFNs) [190, 191]. Type I interferons (IFN-α and IFN-β) are major contributors to combat viral encephalitis, including HSE. Non-immune cells also produce interferons in response to pathogens to regulate the immune system's activity. Studies have proven that overexpression of these cytokines or chemokines could exacerbate brain damage over time. In summary, when HSV-1 infects neurons, innate immunity through IFN signaling prevents the virus from spreading [195].

Adaptive immune responses to HSV-1 and HSV-2 are complicated, involving a variety of factors, including n*uclear factor kappa B* (NFκB) and Janus kinase (JAK)/signal transducer and activator of transcription (STAT) [196]. Interferons could activate IFN stimulated genes (ISGs) that produce antiviral proteins in infected and surrounding cells [196]. The binding of interferons to the IFNα/β receptor activates the JAK-STAT pathway in adjacent cells. This event inhibits viral replication [197]. The innate immune response against HSV is facilitated by plasmacytoid dendritic cells (pDCs) and Natural Killer (NK) cells [196, 198]. Besides interferon, it has been demonstrated that TLR signaling could trigger the expression of proinflammatory cytokines, including pro–IL-1 β leading to neural death [199].

Wang et al. [200] reported that when comparing the expression of cytokines/chemokines in brain tissues from the experimental and control groups, 13 out of 62 components in the array were found to be differently expressed in the experimental group. Notably, all of the differentially expressed proteins (DEPs) were upregulated in the experimental group and belonged to a variety of cytokine families, including the interleukin family (IL-1, IL-2, IL-12f, IL-4, and IL-6), the CCL series (CCL-5, Macrophage inflammatory protein-1 (MIP-1 or CCL-3), and Monocyte chemoattractant protein-1 (MCP-1 or CCL-2)), the CXCL family (keratinocyte-derived cytokine), and the tumor necrosis factor family (TNF-α and soluble tumor necrosis factor receptor- (sTNF-R)). They mentioned that the majority of DEPs were inflammatory proteins that performed important roles in the recruitment and activation of inflammatory

**Table 3. Summary of the meta-analyses.**

| Marker | Source | Subgroup | No. Studies | No. Cases | No. Controls | Meta-analysis | | | Heterogeneity | | | | P-value for subgroup difference |
|---|---|---|---|---|---|---|---|---|---|---|---|---|---|
| | | | | | | SMD | 95% CI | P-value | $I^2$ | $\tau^2$ | Q | P-value[1] | |
| IL-2 | CSF | Autoimmune | 1 | 147 | 35 | NA | NA | NA | NA | NA | NA | NA | NA |
| | | Infectious | 2 | 29 | 48 | 0.87 | -0.89–2.63 | 0.33 | 91% | 1.47 | 11.57 | <0.01 | |
| | | Overall | 3 | 176 | 83 | 0.82 | -0.02–1.66 | 0.05 | 83% | 0.45 | 11.58 | <0.01 | |
| IL-4 | CSF | Autoimmune | 2 | 151 | 41 | 0.50 | -0.89–1.90 | 0.48 | 78% | 0.82 | 4.51 | <0.01 | 0.31 |
| | | Infectious | 3 | 90 | 65 | 2.73 | -1.35–6.80 | 0.19 | 98% | 12.69 | 118.72 | <0.01 | |
| | | Overall | 5 | 241 | 106 | 1.74 | -0.09–3.58 | 0.06 | 97% | 4.15 | 123.24 | <0.01 | |
| IL-6 | CSF | Autoimmune | 7 | 290 | 157 | 1.90 | 0.71–3.09 | <0.01 | 95% | 2.38 | 127.54 | <0.01 | <0.01 |
| | | Infectious | 2 | 50 | 25 | 1.28 | 0.75–1.80 | <0.01 | 0% | 0 | 0 | 0.96 | |
| | | Overall | 9 | 340 | 182 | 1.72 | 0.79–2.66 | <0.01 | 94% | 1.83 | 129.24 | <0.01 | |
| | Serum | Autoimmune | 1 | 16 | 9 | NA | NA | NA | NA | NA | NA | NA | NA |
| | | Infectious | 2 | 95 | 85 | 0.92 | -0.75–2.61 | 0.28 | 91% | 1.35 | 11.1 | <0.01 | |
| | | Overall | 3 | 111 | 94 | 0.56 | -0.41–1.52 | 0.26 | 83% | 0.59 | 11.86 | <0.01 | |
| IL-8 | CSF | Autoimmune | 2 | 151 | 41 | 1.15 | 0.77–1.52 | <0.01 | 0 | 0 | 0.79 | 0.37 | 0.94 |
| | | Infectious | 5 | 190 | 85 | 1.09 | -0.19–2.38 | 0.10 | 94% | 1.99 | 66 | <0.01 | |
| | | Overall | 7 | 341 | 126 | 1.03 | 0.17–1.90 | 0.02 | 92% | 1.19 | 70.63 | <0.01 | |
| IL-10 | CSF | Autoimmune | 5 | 233 | 88 | 2.18 | 0.60–3.77 | <0.01 | 95% | 2.96 | 81.74 | <0.01 | 0.02 |
| | | Infectious | 3 | 61 | 53 | 0.14 | -0.47–0.74 | 0.66 | 52% | 0.14 | 4.13 | 0.13 | |
| | | Overall | 8 | 294 | 141 | 1.31 | 0.36–2.26 | <0.01 | 93% | 1.64 | 95.08 | <0.01 | |
| | Serum | Autoimmune | 1 | 16 | 9 | NA | NA | NA | NA | NA | NA | NA | NA |
| | | Infectious | 2 | 121 | 79 | 0.43 | -0.05–0.92 | 0.08 | 53% | 0.07 | 2.11 | 0.15 | |
| | | Overall | 3 | 137 | 88 | 0.51 | 0.21–0.80 | <0.01 | 7% | 0.01 | 2.16 | 0.34 | |
| IL-17 | CSF | Autoimmune | 4 | 77 | 84 | 0.41 | -0.52–1.34 | 0.38 | 85% | 0.73 | 19.42 | <0.01 | NA |
| | | Infectious | 1 | 13 | 6 | NA | NA | NA | NA | NA | NA | NA | |
| | | Overall | 5 | 90 | 90 | 0.44 | -0.32–1.20 | 0.26 | 80% | 0.47 | 19.51 | <0.01 | |
| CCL2 | CSF | Autoimmune | 2 | 151 | 41 | -0.54 | -0.90--0.18 | <0.01 | 0 | 0 | 0 | 1 | 0.86 |
| | | Infectious | 2 | 50 | 25 | -0.40 | -1.89–1.10 | 0.60 | 85% | 1.00 | 6.85 | <0.01 | |
| | | Overall | 4 | 201 | 66 | -0.53 | -1.08–0.01 | 0.06 | 58% | 0.17 | 7.11 | 0.07 | |
| CXCL9 | CSF | Autoimmune | 1 | 147 | 35 | NA | NA | NA | NA | NA | NA | NA | NA |
| | | Infectious | 2 | 123 | 35 | 3.58 | 3.02–4.14 | <0.01 | 0 | 0 | 0.56 | 0.46 | |
| | | Overall | 3 | 270 | 70 | 2.50 | -0.15–5.16 | 0.06 | 98% | 5.34 | 99.75 | <0.01 | |
| CXCL10 | CSF | Autoimmune | 1 | 147 | 35 | NA | NA | NA | NA | NA | NA | NA | NA |
| | | Infectious | 3 | 57 | 38 | 1.16 | 0.22–2.10 | 0.02 | 74% | 0.50 | 7.72 | 0.02 | |
| | | Overall | 4 | 204 | 73 | 0.86 | 0.16–1.56 | 0.02 | 77% | 0.37 | 13.25 | <0.01 | |
| CXCL10 | Serum | Infectious | 4 | 78 | 51 | 0.12 | -1.27–1.51 | 0.87 | 92% | 1.84 | 37.08 | <0.01 | NA |
| CXCL13 | CSF | Autoimmune | 2 | 163 | 44 | 1.39 | -0.66–3.44 | 0.18 | 92% | 2.01 | 12.23 | <0.01 | NA |
| | | Infectious | 1 | 15 | 8 | NA | NA | NA | NA | NA | NA | NA | |
| | | Overall | 3 | 178 | 52 | 0.91 | -0.23–2.05 | 0.12 | 85% | 0.84 | 13.57 | <0.01 | |
| CXCL13 | Serum | Autoimmune | 1 | 16 | 9 | NA | NA | NA | NA | NA | NA | NA | NA |
| | | Infectious | 2 | 38 | 28 | -0.52 | -2.13–1.10 | 0.53 | 89% | 1.20 | 8.74 | <0.01 | |
| | | Overall | 3 | 54 | 37 | 0.04 | -1.48–1.56 | 0.96 | 91% | 1.62 | 21.38 | <0.01 | |
| TNF-α | CSF | Autoimmune | 3 | 213 | 73 | 1.77 | 0.45–3.09 | <0.01 | 93% | 1.24 | 29.21 | <0.01 | 0.37 |
| | | Infectious | 2 | 102 | 70 | 10.91 | -9.02–30.84 | 0.28 | 98% | 203.47 | 61.13 | <0.01 | |
| | | Overall | 5 | 315 | 143 | 2.61 | 1.21–3.94 | <0.01 | 96% | 1.83 | 90.82 | <0.01 | |
| | Serum | Infectious | 4 | 144 | 106 | 9.06 | 2.85–15.27 | <0.01 | 99% | 38.36 | 317.69 | <0.01 | NA |

*(Continued)*

**Table 3.** (Continued)

| Marker | Source | Subgroup | No. Studies | No. Cases | No. Controls | Meta-analysis | | | Heterogeneity | | | | P-value for subgroup difference |
|---|---|---|---|---|---|---|---|---|---|---|---|---|---|
| | | | | | | SMD | 95% CI | P-value | $I^2$ | $\tau^2$ | Q | P-value[1] | |
| **IFN-γ** | CSF | Autoimmune | 2 | 151 | 41 | 0.35 | -0.01–0.70 | 0.06 | 0 | 0 | 0.4 | 0.53 | 0.19 |
| | | Infectious | 4 | 101 | 93 | 2.06 | -0.50–4.62 | 0.11 | 97% | 6.59 | 117.52 | <0.01 | |
| | | Overall | 6 | 252 | 134 | 1.40 | -0.06–2.85 | 0.06 | 96% | 3.07 | 123.06 | <0.01 | |

[1] P-value for Cochrane Q test

Abbreviations: CSF, cerebrospinal fluid; NA, not applicable

cells. It had not been previously documented that the cytokines IL-1, MIP-1, and sTNF-R were upregulated in HSE [200]. It is generally established that IL-1, regulated upon activation, normal T cells expressed and secreted (RANTES or CCL-5), and KC stimulate the accumulation of inflammatory cells in the affected tissues. MIP-1 and MCP-1 are also known to stimulate the formation of monocyte/macrophage cell lines, whereas IL-2, TNF-α, and IL-12 are known to promote the activation and differentiation of T cells. Furthermore, IL-4 and IL-6 increase B cell activation and differentiation [201, 202]. These elements could also trigger the expression of additional cytokines and probably have a critical role in increasing inflammation and brain tissue damage in acute viral meningitis [203, 204].

The Japanese Encephalitis Virus (JEV) is the most prevalent cause of viral encephalitis in Southeast Asia, transmitted by mosquitoes. After JEV has entered the bloodstream, the viral replication peaks without any cell death or TNF-α release; then, monocytes become activated and differentiate into monocyte-derived dendritic cells (MDDCs) and monocyte-derived macrophages (MDMs) [205]. Pro-inflammatory cytokines such as TNF-α, IL-12, and IL-6 are produced by DCs infected with JEV; however, IL-10 is not produced by macrophages infected with JEV [206]. The microglial cells, which are CNS resident macrophages, could also be infected by JEV. Microglial cells play a critical role in the CNS during the JEV infection because they appear as a virus reservoir [207]. Activated microglia produce pro-inflammatory cytokines, namely TNF-α and IL-6, resulting in neuronal cell death [208]. Moreover, the study of Winter et al. [100], which was a large investigation into cytokines and chemokines for any acute viral encephalitis both in terms of the number of patients and number of investigated parameters, has demonstrated that an unregulated pro-inflammatory response in flavivirus encephalitis can be detrimental. TNF-α, IL-8, and IFN-α have been found to be related to poor outcomes in prior investigations of Japanese encephalitis (JE) patients [29, 54, 209]. In Winter et al.'s study [100], it has been established that TNF-α, IL-8, and IFN-α are critical in the development of JE, as well as the importance of IL-6 and RANTES. In addition, the West Nile virus and other flaviviruses have been shown to harm the CNS in a variety of animal models [210, 211].

Non-infectious, immune-mediated inflammation of the brain parenchyma is called autoimmune encephalitis. It can affect the cortex or deep gray matter, as well as white matter, the meninges, and the spinal cord [212–215]. Cytokines and chemokines in autoimmune encephalitis may give insight into the pathophysiology of autoimmune encephalitis. Studies in patients with NMDAR antibody-associated autoimmune encephalitis have shown an increase in serum IL-2 [172] and CSF IL-6, IL-17, CXCL-10, and IL-1β. Both innate and adaptive immunity, specifically Th1 and Th17, are involved in the secretion of these cytokines [13, 42, 151, 172, 216].

Monocytes and microglia have been found to produce CXCL13 [146]. it is a B cell chemoattractant that was found in the CSF of NMDAR antibody-associated autoimmune encephalitis

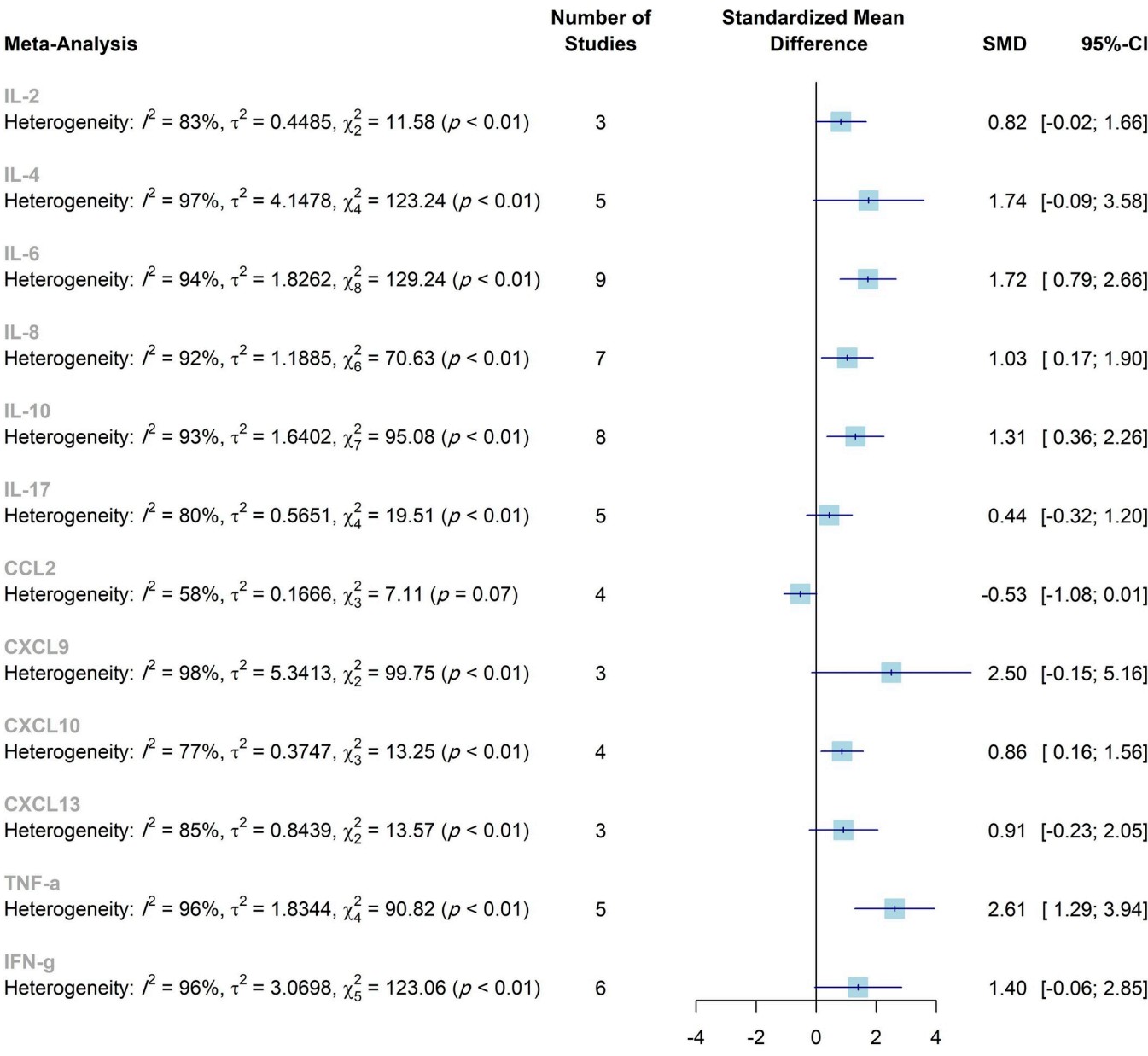

**Fig 7. Summary of the overall effects and heterogeneity of the concentration of the cytokines in the CSF.** SMD, standardized mean difference; CI, confidence interval.

patients. A decrease in CXCL13 levels was associated with a positive outcome after therapy [146]. The levels of BAFF and APRIL in the CSF of autoimmune encephalitis patients were linked to functional results in a study. According to the opposite findings, no increase in CSF levels of BAFF or APRIL was seen in research comparing NMDAR antibody-associated auto-immune encephalitis with viral encephalitis [217, 218].

Interferon-γ, IL-17, IL-12, and IL-23 levels in the CSF of autoimmune encephalitis patients with antibodies to cell surface proteins are greater than those of autoimmune encephalitis patients with antibodies to intracellular antigens [129]. According to recently published research, patients with NMDAR antibody-associated autoimmune encephalitis had increased

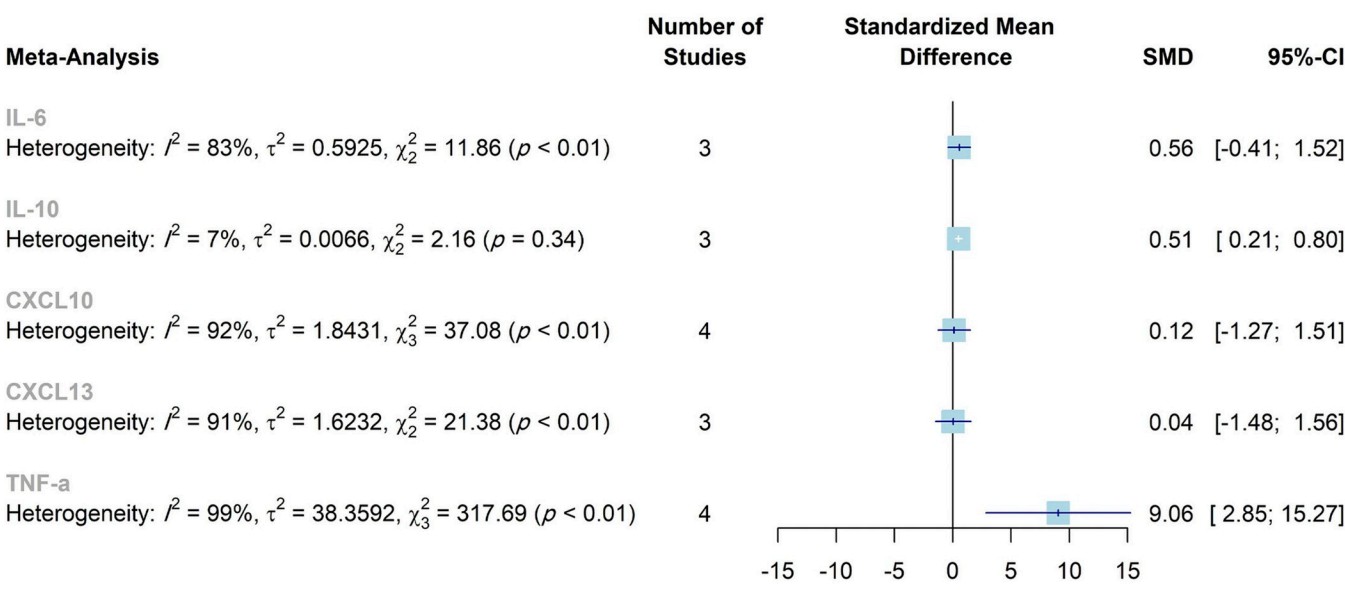

**Fig 8. Summary of the overall effects and heterogeneity of the concentrations of the cytokines in the serum/plasma.** SMD, standardized mean difference; CI, confidence interval.

IL-6, pentraxin-3, CD40L, and IL-17A in their CSF [141]. Patients with new-onset refractory status epilepticus who had autoimmune epilepsy were studied in a study. It was discovered that the CSF had higher IL-6, TNF-α, IL-2, and IL-4 and elevated levels of IL-6 and TNF-α in the peripheral tissues were observed [219]. Eighty-six percent of patients had improvement in seizure activity and a return to normal cytokine levels after receiving therapy with a monoclonal antibody targeting the IL-6 receptor (Tocilizumab) [219].

Our meta-analysis has several limitations. Some of the included cytokines lacked enough original studies to perform the meta-analysis. Despite performing subgroup analysis for the types of encephalitis (infectious or autoimmune), The etiologies of the disease, which may affect the concentration of cytokines in CSF or serum, were not included in the meta-analysis due to insufficient data. Moreover, due to inadequate data, the meta-regression for age, as a possible moderator variable, was not applicable in all cytokines and chemokines. Also, the sample size of most studies was small. Regarding the measurement methods, the studies used different methods and kits, which resulted in heterogeneity. Additionally, the heterogeneity was high in most of the meta-analyses. Sensitivity analysis was conducted to investigate the probable sources of heterogeneity, which remained high in some instances. To the best of our knowledge, this is the first meta-analysis investigating the serum and CSF concentrations of the cytokines in encephalitis. Also, the levels of the cytokines have been compared between autoimmune and infectious encephalitis in subgroup analysis.

## 5. Conclusions

This meta-analysis serves as evidence that encephalitis patients had greater CSF concentrations of IL-6, IL-8, IL-10, CXCL10, and TNF-α when compared to controls. TNF-α also showed increased concentration in serum. Moreover, it was observed that IL-10 had higher levels in autoimmune encephalitis compared to autoimmune encephalitis. Contrastingly, CXCL9 had a higher concentration in infectious encephalitis. Accordingly, the interleukin antagonists could be investigated as a potential adjunctive treatment in encephalitis. The diagnostic and prognostic sensitivity and specificity of these cytokines may also be evaluated in future studies.

Furthermore, the prospect of antiviral medication and other viable treatment methods needs much more extensive exploration to avoid disease progression and the frequently severe sequalae after encephalitis infection.

## Supporting information

**S1 Checklist. Completed PRISMA checklist.**
(PDF)

**S1 Table. Full search strategy for each database.**
(PDF)

**S1 Protocol. Meta-analysis protocol registered in PROSPERO.**
(PDF)

**S1 File. Funnel plots, drapery plot, and leave-one-out analyses for all cytokines.**
(PDF)

**S1 Fig. Summary of serum/plasma meta-analyses.**
(TIF)

**S2 Fig. Summary of CSF meta-analyses.**
(TIF)

## Author Contributions

**Conceptualization:** Alireza Soltani Khaboushan, Mohammad-Taha Pahlevan-Fallahy, Parnian Shobeiri, Antônio L. Teixeira, Nima Rezaei.

**Data curation:** Alireza Soltani Khaboushan, Mohammad-Taha Pahlevan-Fallahy, Parnian Shobeiri.

**Formal analysis:** Alireza Soltani Khaboushan.

**Methodology:** Alireza Soltani Khaboushan, Mohammad-Taha Pahlevan-Fallahy, Antônio L. Teixeira, Nima Rezaei.

**Supervision:** Nima Rezaei.

**Validation:** Antônio L. Teixeira, Nima Rezaei.

**Visualization:** Alireza Soltani Khaboushan.

**Writing – original draft:** Alireza Soltani Khaboushan, Mohammad-Taha Pahlevan-Fallahy, Parnian Shobeiri.

**Writing – review & editing:** Alireza Soltani Khaboushan, Mohammad-Taha Pahlevan-Fallahy, Parnian Shobeiri, Antônio L. Teixeira, Nima Rezaei.

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
