## [Decision Letter · Decision Letter 0]

15 Jul 2022

PONE-D-22-04861Cytokines and chemokines profile in encephalitis patients: a meta-analysisPLOS ONE

Dear Dr. Rezaei,

Thank you for submitting your manuscript to PLOS ONE. After careful consideration, we feel that it has merit but does not fully meet PLOS ONE’s publication criteria as it currently stands. Both reviewers identified significant concerns with the parameters and analysis in this study that need to be addressed. Therefore, we invite you to submit a revised version of the manuscript that addresses the points raised during the review process.  

We look forward to receiving your revised manuscript.

Kind regards,

Karin E. Peterson, Ph.D.

Academic Editor

PLOS ONE

Journal Requirements:

Reviewers' comments:

Reviewer's Responses to Questions

**Comments to the Author**

1. Is the manuscript technically sound, and do the data support the conclusions?

Reviewer #1: Yes

Reviewer #2: Partly

2. Has the statistical analysis been performed appropriately and rigorously?

Reviewer #1: I Don't Know

Reviewer #2: Yes

3. Have the authors made all data underlying the findings in their manuscript fully available?

Reviewer #1: Yes

Reviewer #2: Yes

4. Is the manuscript presented in an intelligible fashion and written in standard English?

Reviewer #1: Yes

Reviewer #2: Yes

5. Review Comments to the Author

Reviewer #1: Dear Authors,

This is a well-written study describing the alterations in the levels of different cytokines in patients with encephalitis. The authors reviewed dozens of articles and selected 23 of them for meta-analysis.

My major concern is that the authors write that 71 studies were eligible for study inclusion, but only 23 were included in the meta-analysis. However, the criteria for selecting 23 out of 71 were not described and should be explained.

Minor remark:

In the abstract the authors write: „Also, these cytokines and chemokine might potentially prevent severe sequelae related to encephalitis infection.”

However, this sentence seems loosely related the manuscript, as the manuscript does not deal with the role of cytokines in the course of encephalitis and its compilations.

Reviewer #2: General Comments-

The authors undertake a meta-analysis of existing literature concerning cytokine/chemokine concentrations in serum/plasma and CSF samples from patients diagnosed with encephalitis compared to in-study controls. In principle, this is a study of interest that could inform diagnostic and therapeutic strategies to deal with encephalitis. However, as is common with meta-analysis studies, the high degree of variability between studies and the generalizations employed (ie. Infectious v. autoimmune encephalitis) lessens confidence in the findings. Despite these drawbacks, with additional selection criteria, subgroup analyses and a clear and inclusive discussion of the limitations of this study, this work could be publishable.

Major concerns-

1) While useful as a starting point, the assumption/generalization that the cytokine/chemokine profile for any given autoimmune encephalitis or encephalitis of infectious origin is going to be similar is flawed and may be diluting important findings. Multiple studies within this meta-analysis describe the etiology of the encephalitis. For example, Japanese encephalitis and NMDA-autoimmune patients are described in multiple studies with overlapping cytokine/chemokines analyzed. The authors should provide additional subgroup analyses that characterize confirmed cases to provide a more accurate cytokine/chemokine profile in those conditions. Conversely, analyzing diagnostically unconfirmed cases as a group may reveal a unique cytokine/chemokine signature and should be untaken.

2) The authors state that (line 163) “Regarding the studies that had investigated more than one type of encephalitis, we pooled the mean and SD of the concentration of these groups”. In line with comment 1), this is inappropriate as written. Were the data pooled regardless of being encephalitis of infectious or autoimmune origin as this would imply? This needs to be clarified. Infectious vs. autoimmune data should be subgrouped and if the etiologic agent/condition is known, the data should be subgrouped again.

3) It does not appear that patient age is considered in these analyses. Several of these studies were performed in pediatric populations and some patient populations contained very old individuals. Cytokine/chemokine levels are differentially regulated at younger and older ages. Statistical tests should be run to ensure that age is not significantly influencing the analysis.

4) The discussion section needs to be largely rewritten. The second and the majority of third paragraphs (line 526-556) read as a textbook synopsis of adaptive and innate immune responses to HSV in the brain and largely do not relate directly to work presented.

Furthermore, little attention is paid to the limitations and caveats of this study, of which there are many. Discussion should include, patient and control populations age, gender, condition (ie. healthy v. nonspecific health conditions), location, immune experience and assay sensitivity.

Minor points-

Line 73-74- “More recently, there has been a growing interest towards autoimmune encephalitis, such as Anti-N-Methyl-D-Aspartate Receptor (NMDAR)”. This is an incomplete sentence.

Line 83- “causative” agents.

Line 95-96- Cytokines are also produced by cells in the tissue. References should be provided for these statements.

Line 103-104- Chemokine are also produced by cells in tissue.

Line 108- Lower case “chemokines”.

Line 123- Wrong protocol is linked.

Line 135- All patients with cancer should be excluded, correct? Tumors for example can change basal cytokine/chemokine levels themselves or induce changes in secretion by other cells.

Line 206- add “in”

Can not see all of tables 1 and 3. Transform to landscape.

Table 2- Should 2: be 0:?

Line 370- add “the”

Line 627-629- Heterogeneity was high in “most” of the meta-analyses.

Line 640- “Accordingly, the interleukin antagonists should be investigated as adjunctive treatment in encephalitis.”- This may be flawed logic as the induced immune response if often critical to resolution of infectious encephalitis. This language should be tempered.

6. PLOS authors have the option to publish the peer review history of their article (what does this mean?). If published, this will include your full peer review and any attached files.

**Do you want your identity to be public for this peer review?** For information about this choice, including consent withdrawal, please see our Privacy Policy.

Reviewer #1: No

Reviewer #2: No

 **********In compliance with data protection regulations, you may request that we remove your personal registration details at any time. (Remove my information/details). Please contact the publication office if you have any questions.

---

## [Author Response · Author response to Decision Letter 0]

30 Jul 2022

Reviewers' comments:

Comments to the Author

Comments From the Reviewer #1:

• Comment:

- Dear Authors,

This is a well-written study describing the alterations in the levels of different cytokines in patients with encephalitis. The authors reviewed dozens of articles and selected 23 of them for meta-analysis.

Response:

- Thank you very much for the time you put into reviewing our manuscript and thank you for your comments. We tried our best to address all your comments, and the detailed answers are in the following sections.

• Major comment:

- My major concern is that the authors write that 71 studies were eligible for study inclusion, but only 23 were included in the meta-analysis. However, the criteria for selecting 23 out of 71 were not described and should be explained.

Response:

- Thank you for your comment and for pointing this out. 71 studies reported data about changes in the concentration of cytokines in encephalitis patients, but only 23 had enough reliable quantitative data to be included in the meta-analysis. Most of the data available in the other studies were not quantitative. We did not have qualitative synthesis, and we only performed meta-analyses in our study. Based on your comment, the section was edited, and more explanation was added to clarify this item.

“Based on the inclusion and exclusion criteria, 71 studies [8, 23-30, 121, 125-185] were considered for this study, of which 23 studies had sufficient quantitative data and were included in the meta-analysis and reported in this study [25, 28-30, 126-128, 132, 135, 141-143, 145, 148, 150, 153, 157, 176, 179-181, 184, 185]. The remaining 48 studies were not included in the meta-analysis because they mostly reported qualitative data or did not contain sufficient quantitative data. Our study does not report qualitative data and only focuses on the meta-analysis of quantitative data.”

• Minor Comment:

- In the abstract, the authors write: “Also, these cytokines and chemokine might potentially prevent severe sequelae related to encephalitis infection.”

However, this sentence seems loosely related the manuscript, as the manuscript does not deal with the role of cytokines in the course of encephalitis and its compilations.

Response:

- Thank you very much for your comment and for pointing this out. We removed this sentence from the abstract.

Comments From the Reviewer #2:

• General Comments:

• Comment

- The authors undertake a meta-analysis of existing literature concerning cytokine/chemokine concentrations in serum/plasma and CSF samples from patients diagnosed with encephalitis compared to in-study controls. In principle, this is a study of interest that could inform diagnostic and therapeutic strategies to deal with encephalitis. However, as is common with meta-analysis studies, the high degree of variability between studies and the generalizations employed (ie. Infectious v. autoimmune encephalitis) lessens confidence in the findings. Despite these drawbacks, with additional selection criteria, subgroup analyses, and a clear and inclusive discussion of the limitations of this study, this work could be publishable.

Response:

- Thank you very much for your insightful and constructive comments. We carefully checked them and tried to address all of them. The detailed answers are in the following sections.

• Comment:

- While useful as a starting point, the assumption/generalization that the cytokine/chemokine profile for any given autoimmune encephalitis or encephalitis of an infectious origin is going to be similar is flawed and may be diluting important findings. Multiple studies within this meta-analysis describe the etiology of encephalitis. For example, Japanese encephalitis and NMDA-autoimmune patients are described in multiple studies with overlapping cytokine/chemokines analyzed. The authors should provide additional subgroup analyses that characterize confirmed cases to provide a more accurate cytokine/chemokine profile in those conditions. Conversely, analyzing diagnostically unconfirmed cases as a group may reveal a unique cytokine/chemokine signature and should be untaken.

Response:

- Thank you very much for your comment and for pointing this out. The number of studies that met our inclusion criteria to be considered for meta-analysis was limited. Due to this limitation, we only were able to conduct subgroup analysis for autoimmune encephalitis or encephalitis of infectious origin. Although, as you mentioned, it is of high importance to consider the etiology of the encephalitis in the meta-analysis, the data were not enough to perform a robust meta-analysis. Moreover, based on the extracted data, different origins of the encephalitides did not have too much difference in concentrations of cytokines and chemokines in most cases. However, based on your comments, this analysis was of high interest if it was possible, and we have added this problem to the limitations of the study, and we will consider it for further studies. The limitations are updated in the discussion part, which is as follows:

“Despite performing subgroup analysis for the types of encephalitis (infectious or autoimmune), The etiologies of the disease, which may affect the concentration of cytokines in CSF or serum, were not included in the meta-analysis due to insufficient data. Moreover, the meta-regression for age, as a possible moderator variable, was not applicable in all cytokines and chemokines due to inadequate data.”

• Comment:

- The authors state that (line 163) “Regarding the studies that had investigated more than one type of encephalitis, we pooled the mean and SD of the concentration of these groups”. In line with comment 1), this is inappropriate as written. Were the data pooled regardless of being encephalitis of infectious or autoimmune origin as this would imply? This needs to be clarified. Infectious vs. autoimmune data should be subgrouped and if the etiologic agent/condition is known, the data should be subgrouped again.

Response:

- Thank you for your delicate comment. We agree with you that this part is not well-stated. When there is more than one group (effect) in a study, the meta-analysis will lead to a unit-of-analysis issue (effects from one study may affect the overall effect more than they should, Cochrane Handbook, 2022, section 6.2.1). The “Cochrane Handbook for Systematic Reviews of Interventions” suggest multiple methods to deal with this problem and has stated that combining similar groups is the recommended method (v5, 2011, section 9.3.1). Other methods are also available such as multi-level meta-analysis or separating groups. Because a small number of the studies reported enough data, we used the combining method for groups with similar types of encephalitis in each study to overcome the unit-of-analysis issue. The combination of effects was performed whenever the types of encephalitis were similar. Moreover, in most cases, the etiology of the encephalitis was also similar. However, as you mentioned, the sentence was a little vague, and we changed and restated it as follows:

“When studies have reported concentrations of cytokines in more than one encephalitis group, if the types of encephalitis were similar (all groups had infectious encephalitis or all groups had autoimmune encephalitis), we pooled the mean and SD of the concentrations in those groups.”

• Comment:

- It does not appear that patient age is considered in these analyses. Several of these studies were performed in pediatric populations and some patient populations contained very old individuals. Cytokine/chemokine levels are differentially regulated at younger and older ages. Statistical tests should be run to ensure that age is not significantly influencing the analysis.

Response:

- Thank you so much for your comment and for pointing this out. Based on your comment, we conducted a meta-regression for “age” when it was possible. The mean or median age of participants of each study has been added to the meta-analysis as a moderator (meta-regression). Meta-regression was performed for cytokines/chemokines with more than two studies that reported mean or median age. The meta-regression process has been added to the method section, and the results of meta-regression for all cytokines/chemokines were added to the results section (age meta-regression for serum TNF-α, CSF CXCL13, CSF CXCL9, serum IL-10, serum IL-6, CSF IL-2 concentrations was not applicable). Moreover, the meta-regression demonstrated that IL-8 CSF concentration, CXCL13 serum concentration, and TNF-α CSF concentration are affected by the age of the studies as a moderator factor. 

Meta-regression for age has been explained in the method section:

“Meta-regression was also performed based on the mean age of the participants of each study if it was possible (more than two studies with reported age were present in the meta-analysis; studies that did not report the mean or median age of the participants were omitted from meta-regression).”

• Comment:

- The discussion section needs to be largely rewritten. The second and the majority of third paragraphs (line 526-556) read as a textbook synopsis of adaptive and innate immune responses to HSV in the brain and largely do not relate directly to work presented.

Furthermore, little attention is paid to the limitations and caveats of this study, of which there are many. Discussion should include, patient and control populations age, gender, condition (ie. healthy v. nonspecific health conditions), location, immune experience and assay sensitivity.

Response:

- We appreciate your comment. We have thoroughly checked and edited the discussion section based on your comments, and the limitations of the study also have been revised and added. The discussion is hugely revised and rewritten, and the limitations have been added. The edited discussion is added to the revised manuscript files.

• Minor Comments:

• Comment:

- Line 73-74- “More recently, there has been a growing interest towards autoimmune encephalitis, such as Anti-N-Methyl-D-Aspartate Receptor (NMDAR)”. This is an incomplete sentence.

Response:

- Thank you for your comment. The revised sentence is as follows: “More recently, a growing number of studies have investigated biomarkers for diagnosis and therapeutic targeting of autoimmune encephalitides, such as Anti-N-Methyl-D-Aspartate Receptor (NMDAR) [2-5].”

• Comment:

- Line 83- “causative” agents.

Response:

- Done.

• Comment:

- Line 95-96- Cytokines are also produced by cells in the tissue. References should be provided for these statements.

Response:

- Thank you for your comment. References number 15-18 have been added.

15. Xie WR, Deng H, Li H, Bowen TL, Strong JA, Zhang JM. Robust increase of cutaneous sensitivity, cytokine production and sympathetic sprouting in rats with localized inflammatory irritation of the spinal ganglia. Neuroscience. 2006;142(3):809-22.

16. DeLeo JA, Colburn RW, Nichols M, Malhotra A. Interleukin-6-mediated hyperalgesia/allodynia and increased spinal IL-6 expression in a rat mononeuropathy model. J Interferon Cytokine Res. 1996;16(9):695-700.

17. Schäfers M, Svensson CI, Sommer C, Sorkin LS. Tumor necrosis factor-alpha induces mechanical allodynia after spinal nerve ligation by activation of p38 MAPK in primary sensory neurons. J Neurosci. 2003;23(7):2517-21.

18. Heijmans-Antonissen C, Wesseldijk F, Munnikes RJ, Huygen FJ, van der Meijden P, Hop WC, et al. Multiplex bead array assay for detection of 25 soluble cytokines in blister fluid of patients with complex regional pain syndrome type 1. Mediators Inflamm. 2006;2006(1):28398.

• Comment:

- Line 103-104- Chemokine are also produced by cells in tissue.

Response:

- Thank you for your comment. This part was edited in the manuscript. 

“Chemokines are a group of cytokines that are released mainly by leukocytes to induce chemotaxis in damaged tissues and attract white blood cells. In addition, chemokines could be secreted by tissue-resident cells, such as neurons and glial cells in the brain [20-22].”

20. Mélik-Parsadaniantz S, Rostène W. Chemokines and neuromodulation. J Neuroimmunol. 2008;198(1):62-8.

21. Ubogu EE, Cossoy MB, Ransohoff RM. The expression and function of chemokines involved in CNS inflammation. Trends Pharmacol Sci. 2006;27(1):48-55.

22. Jaerve A, Müller HW. Chemokines in CNS injury and repair. Cell Tissue Res. 2012;349(1):229-48.

• Comment:

- Line 108- Lower case “chemokines”.

Response:

- Done.

• Comment:

- Line 123- Wrong protocol is linked.

Response:

- Thank you for your comment. The URL has been edited in the manuscript.

https://www.crd.york.ac.uk/prospero/display_record.php?ID=CRD42021289298

• Comment:

- Line 135- All patients with cancer should be excluded, correct? Tumors for example can change basal cytokine/chemokine levels themselves or induce changes in secretion by other cells.

Response:

- Thank you for your comment. We agree with you. Cancer can affect the concentration of the cytokine/chemokine in CSF and serum. Included studies did not contain cancer patients, and paraneoplastic encephalitis patients were the only cancer patients. Based on your comment, we added cancer as an exclusion criterion, but we also keep paraneoplastic encephalitis as an exclusion to also emphasize this group. So, the item has been changed as follows:

“6) All cancer patients and patients with paraneoplastic encephalitis, because of the existence of cancerous tissue or remote neoplasia that might dysregulate cytokines profile.”

• Comment:

- Line 206- add “in”

Response:

- Done.

• Comment:

- Can not see all of tables 1 and 3. Transform to landscape.

Response:

- Done.

• Comment:

- Table 2- Should 2: be 0:?

Response:

- Yes, done.

• Comment:

- Line 370- add “the”

Response:

- Done.

• Comment:

- Line 627-629- Heterogeneity was high in “most” of the meta-analyses.

Response:

- Done.

• Comment:

- Line 640- “Accordingly, the interleukin antagonists should be investigated as adjunctive treatment in encephalitis.”- This may be flawed logic as the induced immune response if often critical to resolution of infectious encephalitis. This language should be tempered.

Response:

- Thank you for your comment. The sentence has been tempered as follows:

“Accordingly, the interleukin antagonists could be investigated as a potential adjunctive treatment in encephalitis.”

---

## [Decision Letter · Decision Letter 1]

18 Aug 2022

Cytokines and chemokines profile in encephalitis patients: a meta-analysis

PONE-D-22-04861R1

Dear Dr. Rezaei,

We’re pleased to inform you that your manuscript has been judged scientifically suitable for publication and will be formally accepted for publication once it meets all outstanding technical requirements.

Kind regards,

Karin E. Peterson, Ph.D.

Academic Editor

PLOS ONE

Additional Editor Comments (optional):

Reviewers' comments:

Reviewer's Responses to Questions

**Comments to the Author**

1. If the authors have adequately addressed your comments raised in a previous round of review and you feel that this manuscript is now acceptable for publication, you may indicate that here to bypass the “Comments to the Author” section, enter your conflict of interest statement in the “Confidential to Editor” section, and submit your "Accept" recommendation.

Reviewer #1: All comments have been addressed

Reviewer #2: All comments have been addressed

2. Is the manuscript technically sound, and do the data support the conclusions?

Reviewer #1: Yes

Reviewer #2: Yes

3. Has the statistical analysis been performed appropriately and rigorously? 

Reviewer #1: Yes

Reviewer #2: Yes

4. Have the authors made all data underlying the findings in their manuscript fully available?

Reviewer #1: Yes

Reviewer #2: Yes

5. Is the manuscript presented in an intelligible fashion and written in standard English?

Reviewer #1: Yes

Reviewer #2: Yes

6. Review Comments to the Author

Reviewer #1: Dear Authors,

The manuscript has been improved substantially. All comments have been addressed.

I have no additional remarks.

Reviewer #2: (No Response)

7. PLOS authors have the option to publish the peer review history of their article (what does this mean?). If published, this will include your full peer review and any attached files.

Reviewer #1: No

Reviewer #2: No

---

## [Editor Report · Acceptance letter]

23 Aug 2022

PONE-D-22-04861R1 

Cytokines and chemokines profile in encephalitis patients: a meta-analysis 

Dear Dr. Rezaei:

I'm pleased to inform you that your manuscript has been deemed suitable for publication in PLOS ONE. Congratulations! Your manuscript is now with our production department. 

Kind regards, 

on behalf of

Dr. Karin E. Peterson 

Academic Editor

PLOS ONE